# Selective conversion of $CO_2$ to isobutane-enriched $C_4$ alkanes over InZrO$_x$-Beta composite catalyst

Han Wang[1,2,3], Sheng Fan[1,2,3], Shujia Guo[1,2], Sen Wang [1] ✉, Zhangfeng Qin [1] ✉, Mei Dong[1], Huaqing Zhu[1], Weibin Fan [1] & Jianguo Wang [1,2] ✉

Direct conversion of $CO_2$ to a single specific hydrocarbon with high selectivity is extremely attractive but very challenging. Herein, by employing an InZrO$_x$-Beta composite catalyst in the $CO_2$ hydrogenation, a high selectivity of 53.4% to butane is achieved in hydrocarbons (CO free) under 315 °C and 3.0 MPa, at a $CO_2$ conversion of 20.4%. Various characterizations and DFT calculation reveal that the generation of methanol-related intermediates by $CO_2$ hydrogenation is closely related to the surface oxygen vacancies of InZrO$_x$, which can be tuned through modulating the preparation methods. In contrast, the three-dimensional 12-ring channels of H-Beta conduces to forming higher methylbenzenes and methylnaphthalenes containing isopropyl side-chain, which favors the transformation of methanol-related intermediates to butane through alkyl side-chain elimination and subsequent methylation and hydrogenation. Moreover, the catalytic stability of InZrO$_x$-Beta in the $CO_2$ hydrogenation is considerably improved by a surface silica protection strategy which can effectively inhibit the indium migration.

The hydrogenation of $CO_2$ to hydrocarbons using green hydrogen is now considered a practical process in tackling the globe warming caused by excessive emission of greenhouse gases[1–3] as well as in recycling $CO_2$ as a carbon feedstock to produce value-added chemicals[4–8]. In this regard, the conversion of $CO_2$ into bulk chemicals including light olefins[9–12] and aromatics[13–16] has attracted special attention. Two routes are now proposed for the $CO_2$ hydrogenation, viz., the modified Fischer-Tropsch synthesis (FTS) one using Fe- or Co-based catalysts[17–21] and the methanol-mediated one over a composite catalyst consisting of metal oxides and zeotypes (OX-ZEO)[11,22]. As the FTS route keeps to the Anderson-Schultz-Flory (ASF) rule for the product distribution, a wide spectrum of hydrocarbons are produced. In contrast, the selectivity to the target olefin and aromatic products can be considerably improved by using the OX-ZEO composite catalyst via the methanol-mediated route which dispenses with the ASF rule[16,23].

Great progress has been made in the hydrogenation of $CO_2$ to $C_2$–$C_3$ alkenes/alkanes in recent years. Through the regulation of surface electronic structure of metal oxides and acidic properties of zeotypes, a high selectivity to ethene (ca. 70%)[24], propene + butene (ca. 65%)[25], or propane (ca. 70%)[26] was achieved for the $CO_2$ hydrogenation. Nonetheless, it is very challenging to achieve a high selectivity to a defined hydrocarbon (in particular a relatively long chain one) through the $CO_2$ hydrogenation, due to the chemical inertia of $CO_2$ and the complexity of C−C coupling reactions. For example, the selective hydrogenation of $CO_2$ into $C_{4+}$ hydrocarbons is rarely reported[27], although the $C_{4+}$ hydrocarbons are highly valuable as clean fuel and solvents. In this regard, butane is widely used as fuel (liquefied gas), refrigerant, spray, and feedstock in chemical synthesis. Traditionally, butane is obtained from petroleum cracking[28]. The direct conversion of $CO_2$ to butane under mild reaction conditions may provide a new strategy for the renewable

[1]State Key Laboratory of Coal Conversion, Institute of Coal Chemistry, Chinese Academy of Sciences, P.O. Box 165 Taiyuan, Shanxi 030001, P. R. China. [2]University of Chinese Academy of Sciences, Beijing 100049, P. R. China. [3]These authors contributed equally: Han Wang, Sheng Fan. ✉ e-mail: wangsen@sxicc.ac.cn; qzhf@sxicc.ac.cn; iccjgw@sxicc.ac.cn

production of $C_{4+}$ chemicals. Unfortunately, current reported catalyst in general gave a rather low selectivity to butane (<30%)[29]. In addition, to improve the practicability of $CO_2$ hydrogenation to demanded hydrocarbons, the formation of $CH_4$ (from $CO_2$ methanation) and CO (from reverse water-gas shift, RWGS) should also be suppressed to the utmost.

For the $CO_2$ hydrogenation over a bifunctional OX-ZEO catalyst, first, the oxide moiety should be highly effective in building methanol-related intermediate. $In_2O_3$ is a promising catalyst component for the $CO_2$ hydrogenation to methanol, showing high activity at a high temperature of 280–330 °C[30–32]. To suppress the formation of $CH_4$ and CO, various dopants were used to regulate the crystal structure and surface electronic properties of $In_2O_3$[33–36]; among them, doping Zr into $In_2O_3$ proved to be rather effective. Frei and co-workers observed that Zr was able to increase the concentration of surface oxygen vacancies of $In_2O_3$, which could then promote the adsorption and activation of $CO_2$ and raise the selectivity to methanol[37]. Chen and co-workers found that the interfacial interaction between $In_2O_3$ and $ZrO_2$ was favorable to the formation of methanol[38]. Similarly, a high methanol space time yield (STY) was observed over $InZrO_x$ in the $CO_2$ hydrogenation[39,40].

Next, the zeotype moiety must work in close cooperation with the oxide moiety to achieve a high yield of specific hydrocarbon. The zeotype moiety is responsible for the successive transformation of the methanol-related intermediates generated on the oxide moiety into hydrocarbons on the acid sites in the confined interspace[41,42]. Naturally, the distribution of products on the OX-ZEO catalysts is closely related to the topology framework and acidity of the zeotype moiety. Various composite catalysts like $InZrO_x$/SSZ-13[26], $In_2O_3$/SAPO-34[35], and $In_2O_3$/ZSM-5[43] were used in the $CO_2$ hydrogenation, which were selective to alkanes ($C_2^0$–$C_4^0$), light olefins ($C_2^=$–$C_4^=$), and liquid fuels, respectively. It implies that higher alkanes/alkenes like butane may request the zeotype moiety a relative large pore channel such as Beta, which can accommodate large reaction intermediates and allow the quick diffusion of large molecular products. In addition, the facile synthesis of Beta zeolite with a wide range of Si/Al ratio makes it attractive as a catalyst component in the conversion of methanol[44]. It is then expected that a bifunctional catalyst composed of Zr-doped $In_2O_3$ and H-Beta may give a high yield of $C_4$ hydrocarbons for the $CO_2$ hydrogenation. However, we have seldom noticed such reports in this regard, despite that the In-based catalysts have been widely used in the $CO_2$ hydrogenation to alcohols and light alkene and alkane mixtures. In addition, Wang and co-workers reported recently that the indium species in the oxide moiety would continuously run off during the reaction process, resulting in a rapid poisoning of the acid sites on the zeotype moiety[45], which may also inhibit the application of $In_2O_3$ in the bifunctional catalyst in practice for the $CO_2$ hydrogenation to hydrocarbons.

Herein, a bifunctional composite catalyst consisting of $InZrO_x$ oxide and H-Beta zeolite was designed and employed in the hydrogenation of $CO_2$; a high selectivity of 53.4% to butane in all hydrocarbons (CO free) is achieved under 315 °C, 3.0 MPa, and a space velocity of 1200 mL $g^{-1}$ $h^{-1}$ ($H_2$/$CO_2$ = 3), at a $CO_2$ conversion of 20.4% and a selectivity of only about 2% to $CH_4$. Combining with various characterization measures and density-functional theory (DFT) calculation, it is revealed that the high selectivity to butane originates from the hydrocarbon pool (HCP) in the H-Beta zeolite filled with bulkier methylbenzenes and methylnaphthalenes, via an alkyl side-chain intermediate. Moreover, a surface silica protection strategy was developed, which can effectively inhibit the phase separation of $InZrO_x$ oxide and the migration of In species into the H-Beta zeolite and then considerably improve the catalytic stability of $InZrO_x$-Beta in the $CO_2$ hydrogenation. These results pave a way for the design of stable In-based catalyst in the $CO_2$ hydrogenation to a specific hydrocarbon product.

## Results

### Textural and structural properties of $InZrO_x$

Supplementary Table 1 and Supplementary Fig. 1 give the textural properties of the $InZrO_x$ oxides prepared by different methods determined by $N_2$ sorption. Apparently, the surface area of three $InZrO_x$ oxides decreases in the order of $InZrO_x$(CP) (73 $m^2$ $g^{-1}$) > $InZrO_x$(SG) (59 $m^2$ $g^{-1}$) > $InZrO_x$(HT) (41 $m^2$ $g^{-1}$); in addition, $InZrO_x$(CP) also displays much larger mesopore volume (0.31 $cm^3$ $g^{-1}$) than $InZrO_x$(SG) (0.07 $cm^3$ $g^{-1}$) and $InZrO_x$(HT) (0.15 $cm^3$ $g^{-1}$).

Figure 1 shows the XRD patterns and TEM images of the $InZrO_x$ oxides prepared by different methods. In the HRTEM images, the (211) and (222) crystal facets, with a lattice spacing of 0.405–0.410 and 0.290–0.291 nm, respectively, are resolved for three $InZrO_x$ oxides (Fig. 1a–c)[10]. Five diffraction peaks at 21.5°, 30.6°, 35.5°, 51.0°, and 60.7° are observed for all three $InZrO_x$ oxides (Fig. 1d), corresponding to the (211), (222), (400), (440) and (622) crystal facets of cubic $In_2O_3$, respectively (JCPDS PDF#06-0416)[42]. Furthermore, the TEM images shown in Supplementary Fig. 2 illustrated that all three $InZrO_x$ oxides are aggregates of spherical nano particles (NPs), with a mean size of 9.76 nm for $InZrO_x$(CP) (prepared by co-precipitation), 9.19 nm for $InZrO_x$(SG) (by sol-gel method), and 25.18 nm for $InZrO_x$(HT) (by hydrothermal method). The STEM-EDX elemental mapping results indicate that the In, Zr, and O elements are uniformly dispersed with each other in the $InZrO_x$ oxides (Fig. 1e–i and Supplementary Figs. 3, 4).

The surface electronic state of various $InZrO_x$ oxides was analyzed by XPS, as shown in Supplementary Fig. 5. The peaks at 444.4 and 451.8 eV in the In 3$d$ XPS spectra correspond to 3$d_{5/2}$ and 3$d_{3/2}$ of $In^{3+}$, respectively, while those at 182.4 and 184.8 eV in the Zr 3$d$ XPS spectra are assigned to 3$d_{5/2}$ and 3$d_{3/2}$ of $Zr^{4+}$, respectively. The O 1$s$ XPS spectra reveal three oxygen species in the $InZrO_x$ oxides (Fig. 2a)[46]; the peaks at 532.6, 531.4, and 529.9 eV are ascribed to the surface hydroxyl groups (–OH), oxygen around the vacancies ($O_{defect}$), and lattice oxygen ($O_{lattice}$), respectively. The deconvolution results indicate that the fraction of surface oxygen around the vacancies (representing the abundance of oxygen vacancies) for three $InZrO_x$ oxides decreases in the order of $InZrO_x$(CP) (36.15%) > $InZrO_x$(SG) (32.39%) > $InZrO_x$(HT) (27.65%). In addition, the same sequence of three oxides by the surface oxygen vacancy concentration is manifested by the in situ O 1$s$ XPS results (Supplementary Fig. 6), viz., $InZrO_x$(CP) (43.47%) > $InZrO_x$(SG) (36.75%) > $InZrO_x$(HT) (31.76%). The in situ O 1$s$ XPS gives higher surface oxygen vacancy concentrations than the ex situ one, which is ascribed to the fact that more surface oxygen defects are formed due to the elimination of certain surface oxygen atoms by reduction in the in situ $H_2$-containing atmosphere, in agreement with previous works[24,47].

Figure 2b shows the $H_2$-TPR profiles of various $InZrO_x$ oxides. The high temperature peak (centered around 650 °C) is attributed to the reduction of bulk $In_2O_3$, whereas the low temperature one (150–250 °C) is due to the annihilation of surface oxygen from $In_2O_3$[47]. Apparently, $InZrO_x$(CP) displays a more intense $H_2$ consumption peak (19 $\mu$mol $g^{-1}$) attributed to the removal of surface oxygen at a lower temperature (160 °C), in comparison with $InZrO_x$(SG) (18 $\mu$mol $g^{-1}$, at 180 °C) and $InZrO_x$(HT) (16 $\mu$mol $g^{-1}$, at 220 °C), indicating that $InZrO_x$(CP) has more surface defects (oxygen vacancies), agreeing well with above O 1$s$ XPS results. The Rietveld refinement of the in situ XRD patterns (Supplementary Fig. 7) indicates that the cell volume of $InZrO_x$ decreases after $H_2$ reduction, due to the release of surface oxygen atoms[47]. Interestingly, as shown in Fig. 2c, $InZrO_x$(CP) also displays the highest cell shrinkage ratio than $InZrO_x$(SG) and $InZrO_x$(HT), according well with the more surface defects of $InZrO_x$(CP). Moreover, since the onset temperature for the reduction of bulk $In_2O_3$ to metallic indium species is only around 315 °C, it is conceivable that certain metallic indium species may be generated on the surface of $InZrO_x$ oxide during the reduction and subsequent reaction processes. These metallic indium species may easily migrate

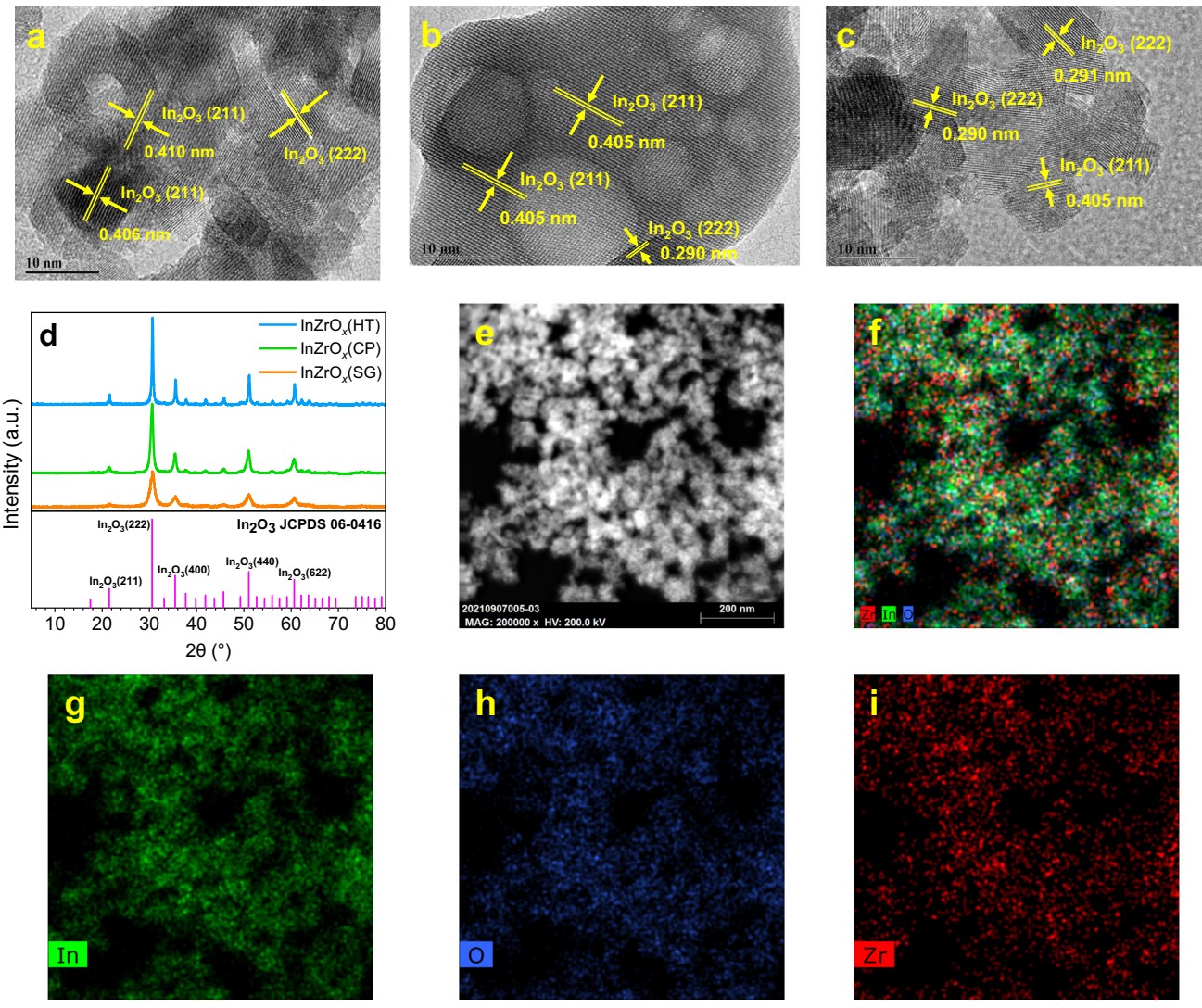

**Fig. 1 | Structure of various InZrO$_x$ oxides. a–c** HRTEM images of InZrO$_x$(CP) (**a**), InZrO$_x$(HT) (**b**), and InZrO$_x$(SG) (**c**); **d** XRD patterns of InZrO$_x$ prepared by different methods; **e–i** STEM-EDX elemental mapping of InZrO$_x$(CP).

to the zeolite moiety and then poison the acid sites of zeolite component[45,47]; this will be discussed in detail in the following section.

Figure 2d displays the CO$_2$-TPD profiles of various InZrO$_x$ oxides. The peak below 150 °C represents the physical adsorption of CO$_2$, whereas that at above 300 °C is ascribed to the chemical adsorption of CO$_2$ on the surface oxygen defects[42]. As expected, InZrO$_x$(CP) exhibits larger CO$_2$ desorption amount at above 300 °C, followed by InZrO$_x$(SG) and InZrO$_x$(HT). It further evinces that CO$_2$ adsorption is enhanced on InZrO$_x$(CP) (prepared by co-precipitation) with abundant oxygen vacancies.

### Properties of H-Beta zeolites

As illustrated in Supplementary Fig. 8a, b, the H-Beta, H-SSZ-13 and USY zeolites used in this work have a typical BEA, CHA and FAU topological framework, respectively, with high crystallinity. N$_2$ sorption results (Supplementary Fig. 8c and Supplementary Table 2) indicate that various H-Beta zeolites with different Si/Al ratios (20–100) are similar in their textural properties. The SEM images shown in Supplementary Fig. 9 display that the H-Beta zeolites has a particle size of around 1 μm. In contrast, the NH$_3$-TPD results given in Supplementary Table 2 and Supplementary Fig. 8d indicate that both the weak acid and strong acid contents of H-Beta decrease with the increase of the Si/Al ratio.

### Catalytic performance of InZrO$_x$-Beta in the CO$_2$ hydrogenation

The InZrO$_x$ oxides alone were first tested for the hydrogenation of CO$_2$ under 315 °C, 3.0 MPa, and with a space velocity (SV) of 2400 mL g$^{-1}$ h$^{-1}$ and H$_2$/CO$_2$ ratio of 3 in the feed. As shown in Supplementary Fig. 10, InZrO$_x$(CP) prepared by co-precipitation exhibits a higher CO$_2$ conversion (18.7%) and a higher methanol space time yield (STY) (0.012 mol kg$^{-1}$ h$^{-1}$) than InZrO$_x$(SG) prepared by sol-gel method (17.3% and 0.011 mol kg$^{-1}$ h$^{-1}$, respectively) and InZrO$_x$(HT) by hydrothermal method (11.2% and 0.003 mol kg$^{-1}$ h$^{-1}$, respectively).

Next, the catalytic performance of granule-mixed bifunctional InZrO$_x$-Beta catalyst (0.3 g InZrO$_x$ + 0.3 g H-Beta) was evaluated in the hydrogenation of CO$_2$ under 315 °C, 3.0 MPa, a SV of 1200 mL g$^{-1}$ h$^{-1}$, and an H$_2$/CO$_2$ ratio of 3 in the feed. As shown in Fig. 3a, over InZrO$_x$-Beta(40), butane is the dominant hydrocarbon product. In particular, InZrO$_x$(CP)-Beta(40) exhibits a selectivity of 53.4% to butane in the hydrocarbon products, at a CO$_2$ conversion of 20.4% and a selectivity of 54.9% to CO. In contrast, over InZrO$_x$(SG)-Beta(40), the selectivity to butane in the hydrocarbon products, CO$_2$ conversion, and selectivity to CO are 52.9%, 18.7%, and 53.0%, respectively, whereas over InZrO$_x$(HT)-Beta(40), the CO$_2$ conversion decreases to 12.6%, whilst the selectivity to butane decreases to 34.7%, accompanied by the formation of more C$_2$–C$_3$ components. It seems that InZrO$_x$(CP) with abundant oxygen vacancies exhibits high activity in the CO$_2$

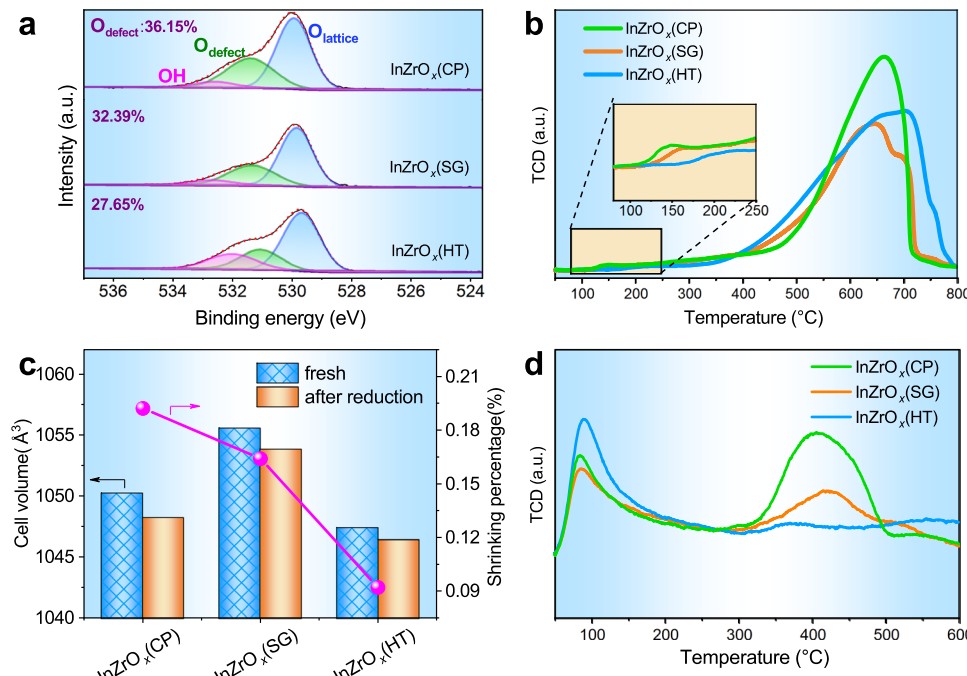

**Fig. 2 | Surface electronic state and reduction behavior of InZrOₓ. a** O 1s XPS spectra of various InZrOₓ oxides, in which the abundance of surface oxygen vacancies ($O_{defect}$, calculated as $I_{defect}/(I_{lattice} + I_{defect})$) are labeled ($I_{defect}$ and $I_{lattice}$ represent peak area of defect and lattice oxygen, respectively)[42]. **b** H₂-TPR profiles of various InZrOₓ oxides (The shadow insert is the enlargement of H₂-TPR profiles in the temperature range of 100–250 °C). **c** Cell volumes of various fresh InZrOₓ oxides compared to that of H₂-reduced ones. **d** CO₂-TPD profiles of various InZrOₓ oxides.

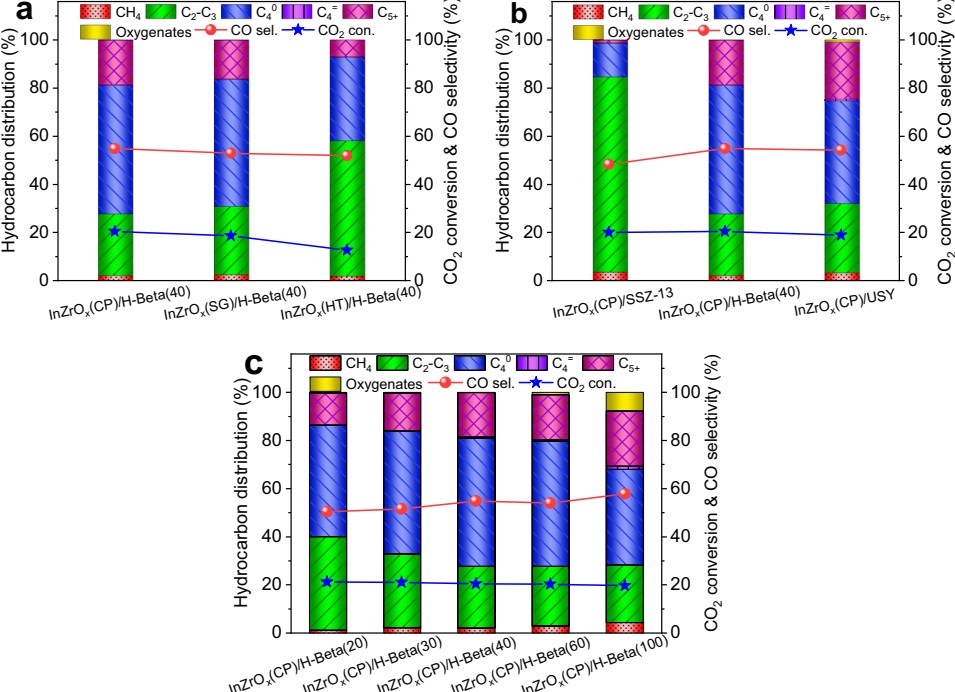

**Fig. 3 | Performance of various composite catalysts in the CO₂ hydrogenation. a**–**c** CO₂ conversion and product distribution for the CO₂ hydrogenation over various InZrOₓ-Beta(40) composite catalysts (**a**); over InZrOₓ(CP) composed with different zeolites (**b**); over InZrOₓ(CP) composed with H-Beta of different Si/Al ratios (**c**). Reaction conditions: 315 °C, 3.0 MPa, SV = 1200 mL g⁻¹ h⁻¹, and H₂/CO₂ = 3.

hydrogenation to methanol; when composed with H-Beta(40), the InZrOₓ(CP)-Beta(40) composite catalyst also displays excellent performance in the tandem transformation of CO₂ to butane.

Besides the oxide moiety, the yield and spectrum of hydrocarbon products for the CO₂ hydrogenation are also closely related to the topology framework and acidic properties of the zeotype moiety in the bifunctional composite catalyst. As shown in Fig. 3b, butane appears as the main hydrocarbon product over InZrOₓ(CP)-Beta, whereas more C₂–C₃ and C₅₊ hydrocarbons are generated over InZrOₓ(CP)-SSZ-13 and InZrOₓ(CP)-USY with smaller windows and larger cavities. It

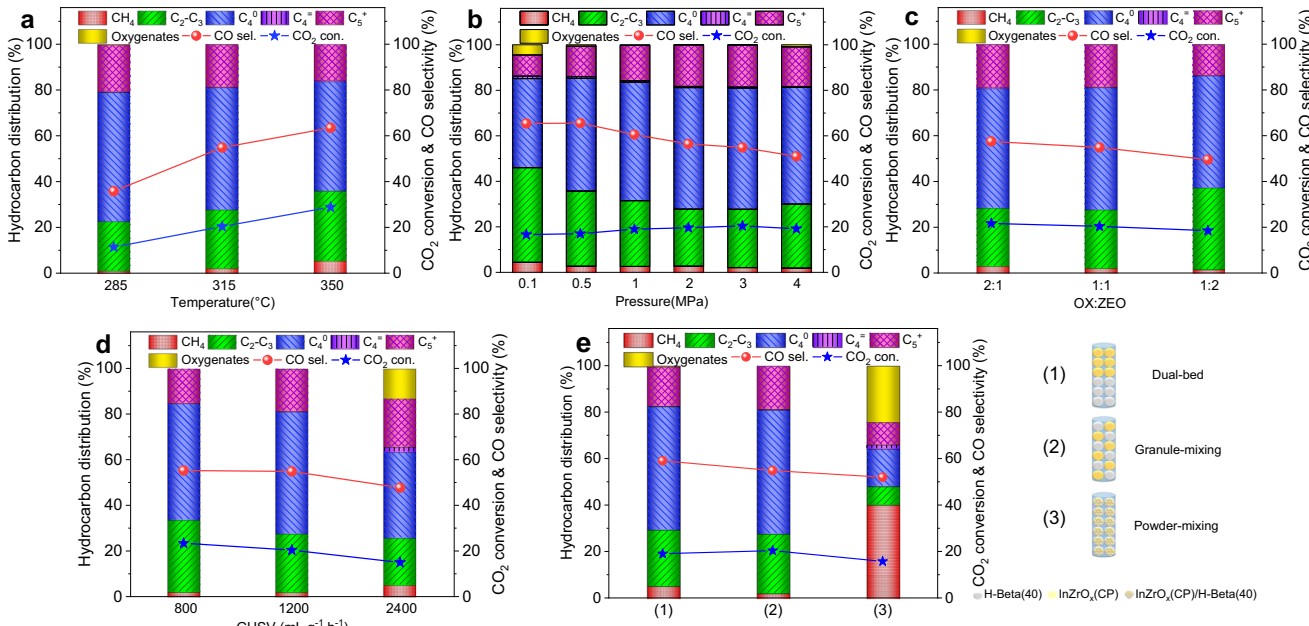

**Fig. 4 | Optimization of reaction conditions for the CO₂ hydrogenation. a–e** CO₂ conversion and product distribution for CO₂ hydrogenation over the InZrO$_x$(CP)-Beta(40) composite catalyst at different temperatures (**a**), pressures (**b**), oxide/ zeolite mass ratios (**c**), space velocities (**d**), and different composing manners of two moieties (**e**). Basic reaction conditions: 315 °C, 3.0 MPa, space velocity of 1200 mL g⁻¹ h⁻¹, and H₂/CO₂ ratio of 3 in the feed.

indicates that the 3D 12-ring channels of H-Beta is pertinent for the transformation of methanol-related intermediates (generated from CO₂ hydrogenation over the oxide moiety) into butane.

Moreover, the product distribution for the CO₂ hydrogenation over InZrO$_x$(CP)-Beta is associated with the acid density of the H-Beta zeolite moiety. As shown in Fig. 3c, with an increase of the Si/Al ratio of H-Beta from 20 to 100, the selectivity to C₂–C₃ components decreases, whereas the fraction of C₅₊ components in the product increases, whilst the selectivity to C₄ hydrocarbons (dominated by butane) achieves the maximum value at a Si/Al ratio of 40. A higher Si/Al ratio (viz., less acid sites in H-Beta) may weaken the capability of alkenes hydrogenation and then yield more alkenes. In particular, InZrO$_x$(CP)-Beta(100) displays even an incomplete conversion of methanol due to its rather low acid density. In contrast, excessive acid sites in H-Beta zeolite (e.g. Beta(20), with a very low Si/Al ratio) conduce to the cracking of long-chain hydrocarbons and then generate more small alkenes/alkanes[48]. Therefore, the selectivity to butane shows a volcanic curve with the Si/Al ratio of H-Beta and gets to the maximum value of 53.4% over InZrO$_x$(CP)-Beta(40). Notably, isobutane accounts for 86.5% of two butane isomers.

As water is a co-product in the hydrogenation of CO₂, which acts also vividly in the competitive reverse water-gas (RWGS) reaction[5,6], the possible role of water in the reaction process was further evaluated by adding different proportions of water into the H₂ and CO₂ feed. As shown in Supplementary Fig. 11a, when water is introduced into the reaction system after about 16 h, the CO₂ conversion decreases from 19.1 to 8.9%, along with the attenuation of the selectivity to CO from 54.7 to 34.0%. Such a phenomenon becomes more evident when the proportion of water in the feed increases from 7.5 to 15.1% (Supplementary Fig. 11b), where the CO₂ conversion and selectivity to CO decreases considerably to 7.3% and 26.1%, respectively. This can be explained by the fact that more water in the reaction mixture can effectively counteract the RWGS reaction (CO₂ + H₂ = CO + H₂O), leading to the decline of the CO₂ conversion and the selectivity to CO[5,6,9]. In addition, water molecules may also compete for the active adsorption sites with the reactants, which may also abate the conversion of CO₂ to hydrocarbons.

It is noteworthy that the conversion of CO₂ and selectivity to CO are both spontaneously rejuvenated, when the co-feeding water is cut off (Supplementary Fig. 11c). However, the CO₂ conversion cannot be fully recovered to the original value, implying that co-feeding water has certain impact on the catalytic activity of the InZrO$_x$ oxide. The XRD patterns and TEM images of the spent catalysts shown in Supplementary Fig. 12 indicate that after the CO₂ hydrogenation with co-feeding water, the particle size of InZrO$_x$ oxide increases considerably, along with the decrease of surface area and pore volume (Supplementary Fig. 13a and Supplementary Table 3). This leads to a decrease in the surface oxygen vacancies concentration, which can weaken the CO₂ adsorption capacity, as indicated by the O 1s XPS and CO₂-TPD results (Supplementary Fig. 13b, c), although the In, Zr and O elements are still uniformly dispersed with each other in the InZrO$_x$ oxide (Supplementary Fig. 14). Nevertheless, the selectivity to butane changes very little during the reaction. This is ascribed to the fact that the crystal structure, morphology, and particle size of H-Beta zeolite, which determine the manner for the formation of hydrocarbons from the methanol-related intermediates, are well maintained during the CO₂ hydrogenation with co-feeding different contents of water (Supplementary Fig. 15).

## Optimization of reaction conditions

The reaction temperature has a great influence on the product yield and spectrum for the hydrogenation of CO₂ over InZrO$_x$(CP)-Beta(40). As shown in Fig. 4a, with the increase of temperature from 285 to 350 °C, as expected, the CO₂ conversion increases from 11.4% to 28.8%, despite that more CO are produced due to the promotion of the reverse water-gas shift (RWGS) reaction at a higher temperature[5,6]. Meanwhile, the selectivity to butane and C₅₊ decreases gradually with the increase of temperature, accompanied by the formation of more C₁–C₃ products due to the aggravated cracking of long-chain hydrocarbons, as also observed by Ding and co-workers[49].

In contrast, an increase of the reaction pressure from 0.1 to 3.0 MPa (at 315 °C) elevates the CO₂ conversion from 16.6 to 20.4%, but decreases the selectivity to CO from 65.4 to 54.9%, as shown in Fig. 4b. Meanwhile, the selectivity to butane is raised from 39.1 to 53.4%,

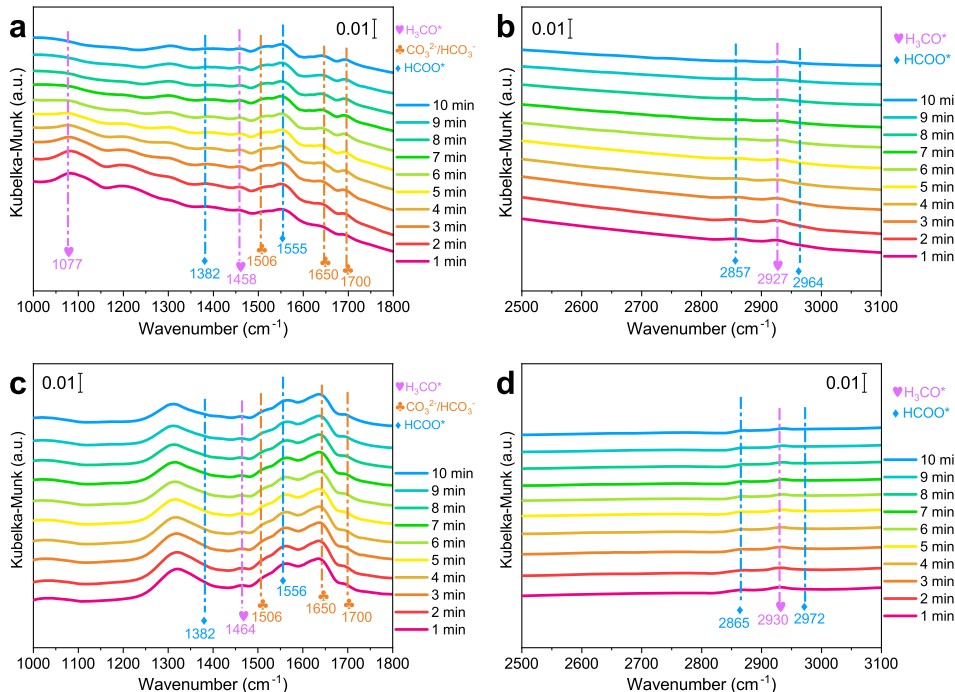

**Fig. 5 | In situ DRIFT spectra. a–d** In situ DRIFT spectra of InZrO$_x$(CP) (**a, b**) and InZrO$_x$(HT) (**c, d**) in CO$_2$ hydrogenation. The spectra were collected every 1 min up to 10 min after pretreating the sample under Ar atmosphere (30 mL min$^{-1}$) for 0.5 h at 200 °C and 0.1 MPa and purging with the H$_2$ and CO$_2$ mixture (40 mL min$^{-1}$, H$_2$/CO$_2$ = 3).

accompanied by a gradual decline of the selectivity to alkenes, as a higher hydrogen pressure conduces to the hydrogenation of alkenes. However, a further increase of pressure to 4.0 MPa leads to the formation of more C$_1$–C$_3$ alkanes, at the expense of butane and C$_{5+}$ hydrocarbons.

The effect of the oxide/zeolite mass ratio in the InZrO$_x$(CP)-Beta(40) composite on the CO$_2$ hydrogenation was also considered, as demonstrated in Fig. 4c. With a decrease of the oxide/zeolite mass ratio from 2 to 1/2, the CO$_2$ conversion and selectivity to CO decrease from 21.7% and 57.5% to 18.5% and 49.6%, respectively, while the highest selectivity to butane (53.4%) is achieved at an oxide/zeolite mass ratio of 1.

The space velocity and H$_2$/CO$_2$ ratio in the feed also have certain influence on the CO$_2$ hydrogenation. A decrease of the space velocity from 2400 to 800 mL g$^{-1}$ h$^{-1}$ makes the CO$_2$ conversion increase from 14.9 to 23.4%, despite that the formation of CO is also promoted (Fig. 4d), whilst the selectivity to butane gets the maximum value of 53.4% at a space velocity of 1200 mL g$^{-1}$ h$^{-1}$. In addition, elevating the H$_2$/CO$_2$ ratio from 3 to 6 in the feed raises the CO$_2$ conversion from 20.4 to 25.6% and decreases the selectivity to CO slightly from 54.9 to 51.2%, while it has little influence on the selectivity to butane, as shown in Supplementary Fig. 16.

In addition, the manner used to compose the InZrO$_x$-Beta bifunctional catalysts also displays a significant influence on the CO$_2$ hydrogenation. As shown in Fig. 4e, a decrease in the contact distance between InZrO$_x$(CP) and H-Beta(40) from dual-bed filling to granule stacking makes the CO$_2$ conversion and selectivity to butane slightly increase from 19.1% and 53.1% to 20.4% and 53.4%, respectively. However, the composite catalyst prepared by powder mixing of the two moieties just produces large amounts of methane (40.2%) and methanol (24.2%), due to the rapid deactivation of H-Beta zeolite. Similar phenomenon was also observed by Wang and co-workers[45]; that is, the In species may easily migrate from the metal oxide moiety to the zeotype moiety when the two moieties are in very close contact, resulting in the rapid passivation of the acid sites in the zeotype moiety

that are responsible for the successive transformation of methanol-related intermediates to hydrocarbons.

On all accounts, the bifunctional InZrO$_x$(CP)-Beta(40) composite catalyst composed by granule stacking with equal mass of oxide and zeolite exhibits excellent performance in the selective hydrogenation of CO$_2$ to butane. Under 315 °C, 3.0 MPa, a space velocity of 1200 mL g$^{-1}$ h$^{-1}$, and an H$_2$/CO$_2$ ratio of 3 in the feed, a high selectivity of 53.4% to butane in all hydrocarbons is achieved, at a CO$_2$ conversion of 20.4% and a selectivity of only about 2% to CH$_4$.

### Reaction mechanism of the CO$_2$ hydrogenation to butane

To reveal the reaction mechanism, in situ DRIFTs were first performed for the CO$_2$ hydrogenation to methanol over the InZrO$_x$(CP), InZrO$_x$(SG), and InZrO$_x$(HT) oxides, as shown in Fig. 5a–d and Supplementary Fig. 17. After introducing H$_2$ and CO$_2$ for reaction over InZrO$_x$(CP) for 1 min, the vibration bands attributed to the carbonate (CO$_3^{2-}$/HCO$_3^-$) species appear at 1506, 1650, and 1700 cm$^{-1}$ [50–52], belonging to the signals of activated CO$_2$ (Fig. 5a, b). Besides, the characteristic peaks assigned to the formate (HCOO*) species at 1382, 1555, 2857, and 2964 cm$^{-1}$ are quickly observed[15,53]. Meanwhile, the peaks at around 1077, 1458, and 2927 cm$^{-1}$ ascribed to the methoxy (H$_3$CO*) species are simultaneously detected[15,53,54]. With the proceeding of the reaction process, the peak intensity of HCOO* and H$_3$CO* species decreases gradually, as they are hydrogenated to methanol. Such phenomena confirm that formate and methoxy are crucial intermediates for methanol formation in the CO$_2$ hydrogenation over InZrO$_x$(CP), in line with the previous reports[15,53]. In comparison with InZrO$_x$(CP), InZrO$_x$(SG) and InZrO$_x$(HT) also show the characteristic peaks of the formate and methoxy species, but in a much lower intensity (Fig. 5c, d and Supplementary Fig. 17), corresponding to their less oxygen vacancies and poorer catalytic performance in the CO$_2$ hydrogenation.

The methanol-related intermediates generated on the InZrO$_x$ oxide are further transformed into hydrocarbons on the acid sites of the H-Beta zeolite, which can be consolidated by the control

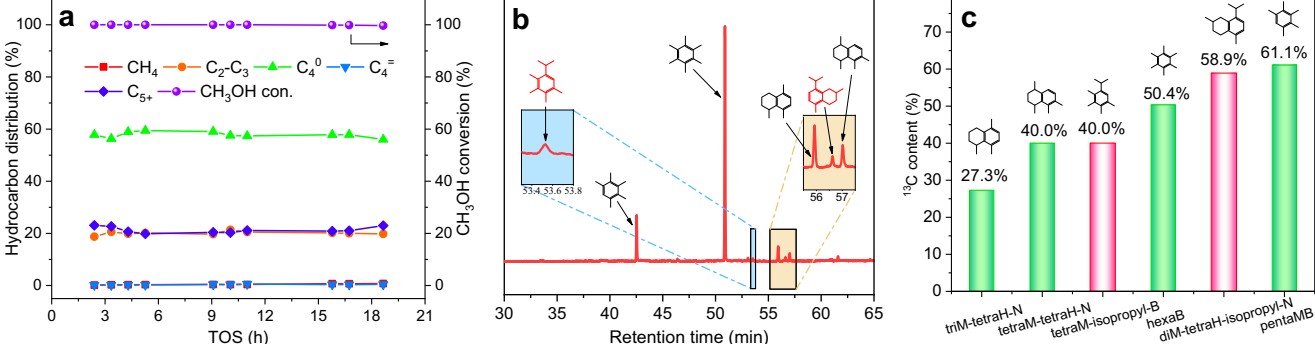

**Fig. 6 | Probe experiments for methanol conversion. a** Methanol conversion and product distribution for MTH in $H_2$ atmosphere over H-Beta(40) zeolite with a methanol WHSV of 0.05 h$^{-1}$. **b** GC-MS chromatograms of residual species in H-Beta(40) after the MTH reaction in $H_2$ atmosphere with a methanol WHSV of 0.05 h$^{-1}$ (The shadow inserts are the enlargement of GC-MS chromatograms at specified retention times). **c** $^{13}$C content of the confined organic species in H-Beta(40), obtained from $^{12}$C/$^{13}$C methanol switching experiment.

experiments for methanol conversion (viz., methanol to hydrocarbons, MTH) in the $H_2$ atmosphere over the H-Beta(40) zeolite. As shown in Fig. 6a, butane is the dominant product in MTH; in addition, the selectivity to butane (58.0%) here in MTH over H-Beta(40) is highly comparable to that in the $CO_2$ hydrogenation over InZrO$_x$-Beta(40) (53.4%).

After the MTH reaction, the residual species in the H-Beta zeolite were extracted and detected by GC-MS[55]. As demonstrated in Fig. 6b, higher polymethylbenzens (polyMBs, e.g., pentaMB and hexaMB) are found to be the dominant hydrocarbon pool (HCP) species generated in the methanol conversion process over the H-Beta zeolite, which could quickly grow up into aromatics containing isopropyl side-chain via continuous methylation reactions. It is known that these polyMBs are responsible for the propene formation via the side-chain aromatic-based cycle[56,57]. After the elimination of isopropyl side-chain, the generated propene can be further converted to butane through methylation and subsequent hydrogenation reactions. In fact, certain polyMBs and polymethylnaphthalenes (polyMNs) with isopropyl side-chain (e.g. 1-isopropyl-2,3,4,5-tetramethylbenzene (isopropyl-MB) and 8-isopropyl-2,5-dimethyl-1,2,3,4-tetrahydronaphthalene (isopropyl-MN)) are simultaneously detected by GC-MS (Fig. 6b). Meanwhile, in the $^{12}$C/$^{13}$C-methanol isotope switching experiments, these higher polyMBs and even polyMNs were labeled by a high content of $^{13}$C (e.g. 61.1% of hexaMB and 58.9% of isopropyl-MN), according well with their high activity in MTH (Fig. 6c).

Furthermore, as expected, these polyMBs and polyMNs as the HCP species are also detected on the spent InZrO$_x$(CP)-Beta(40) catalyst after the $CO_2$ hydrogenation test, as shown in Supplementary Fig. 18. In contrast, the content of polyMBs and polyMNs containing isopropyl side-chain on the spent InZrO$_x$(HT)-Beta(40) catalyst is rather lower than that on the spent InZrO$_x$(CP)-Beta(40) catalyst (Supplementary Fig. 19), corresponding to the rather lower activity of InZrO$_x$(HT) in the $CO_2$ hydrogenation to methanol; that is, as less methanol-related intermediates are formed on the InZrO$_x$(HT) moiety with poorer activity in the $CO_2$ hydrogenation, the formation and growth of higher polyMBs as the HCP species for MTH are also greatly restricted in the H-Beta moiety, leading to the lower yield of butane over the InZrO$_x$(HT)-Beta composite catalyst (Fig. 3a).

The reaction kinetics for butane formation was further investigated by DFT calculation. As shown in Fig. 7 and Supplementary Fig. 20, isopropyl-MB and isopropyl-MN are first protonated to the corresponding carbocations, with a free energy barrier of 79 and 69 kJ mol$^{-1}$, respectively. The elimination of isopropyl group from these two carbocations to form the propoxy species on the acid sites requires a low free energy barrier of 69 and 47 kJ mol$^{-1}$, with a high rate constant of $8.76 \times 10^6$ and $8.45 \times 10^8$ s$^{-1}$, respectively, similar to those

reported previously[56]. Propene is then obtained from the deprotonation of propoxy with a free energy barrier of 70 kJ mol$^{-1}$ and a rate constant of $8.20 \times 10^6$ s$^{-1}$. Further conversion of propene to butoxy and then to iso-butane via methylation, isomerization, and hydrogenation reactions needs to overcome a free energy barrier of 124, 48, and 47 kJ mol$^{-1}$, respectively.

In addition, as given by the calculated free energy surface, the overall free energy height for the butane formation over H-Beta is only 115 kJ mol$^{-1}$, with an overall reaction free energy of −104 kJ mol$^{-1}$. All these evince that butane can be generated easily from the methanol-related intermediates via the side-chain route of aromatic-based cycle in the H-Beta zeolite.

On all accounts, the $CO_2$ hydrogenation over a bifunctional catalyst relies on both the oxide and the zeotype moieties. For the InZrO$_x$-Beta composite, the InZrO$_x$ moiety is responsible for the conversion of $CO_2$ to the methanol-related intermediates (similar to methanol synthesis), whereas the H-Beta moiety is accountable to the subsequent transformation of methanol-related intermediates into hydrocarbons (similar to methanol to hydrocarbons (MTH), via the hydrocarbon pool mechanism). As a result, the conversion of $CO_2$ is mainly related to the InZrO$_x$ moiety; InZrO$_x$(CP) prepared by co-precipitation has abundant oxygen vacancies and great capacity for the $CO_2$ hydrogenation to methanol. In contrast, the product distribution is mainly associated with the framework topology and acidity properties of the zeotype moiety. The H-Beta zeolite of 3D 12-ring channels with moderate acidity (Si/Al = 40) is appropriate for the construction of HCP containing abundant higher polyMBs and polyMNs as well as the production of butane via the aromatic-based cycle of HCP mechanism through the isopropyl side-chain elimination and subsequent methylation and hydrogenation reactions. In addition, the granule-mixing manner used to compose the InZrO$_x$-Beta bifunctional catalyst can realize a pertinent contact between two moieties and then achieve a prominent coupling of two reaction steps (viz., $CO_2$ hydrogenation to methanol and MTH).

## Surface silica modification to restrict the in migration

The degeneration of either the metal oxide moiety or the zeotype moiety can deactivate the whole bifunctional OX-ZEO catalyst system in the hydrogenation of $CO_2$ to hydrocarbons[58,59]. The degeneration of metal oxide often causes a rapid decrease in the $CO_2$ conversion, as the adsorption and activation of $CO_2$ are mainly performed on the surface of metal oxide[37,43]. As for the acidic zeolite, it catalyzes the subsequent transformation of the methanol-related intermediates previously generated on the oxide moiety into hydrocarbons; the rapid increase in the selectivity to unconverted methanol is an important sign for the deactivation of the zeolite component[45,60,61]. For the In-based

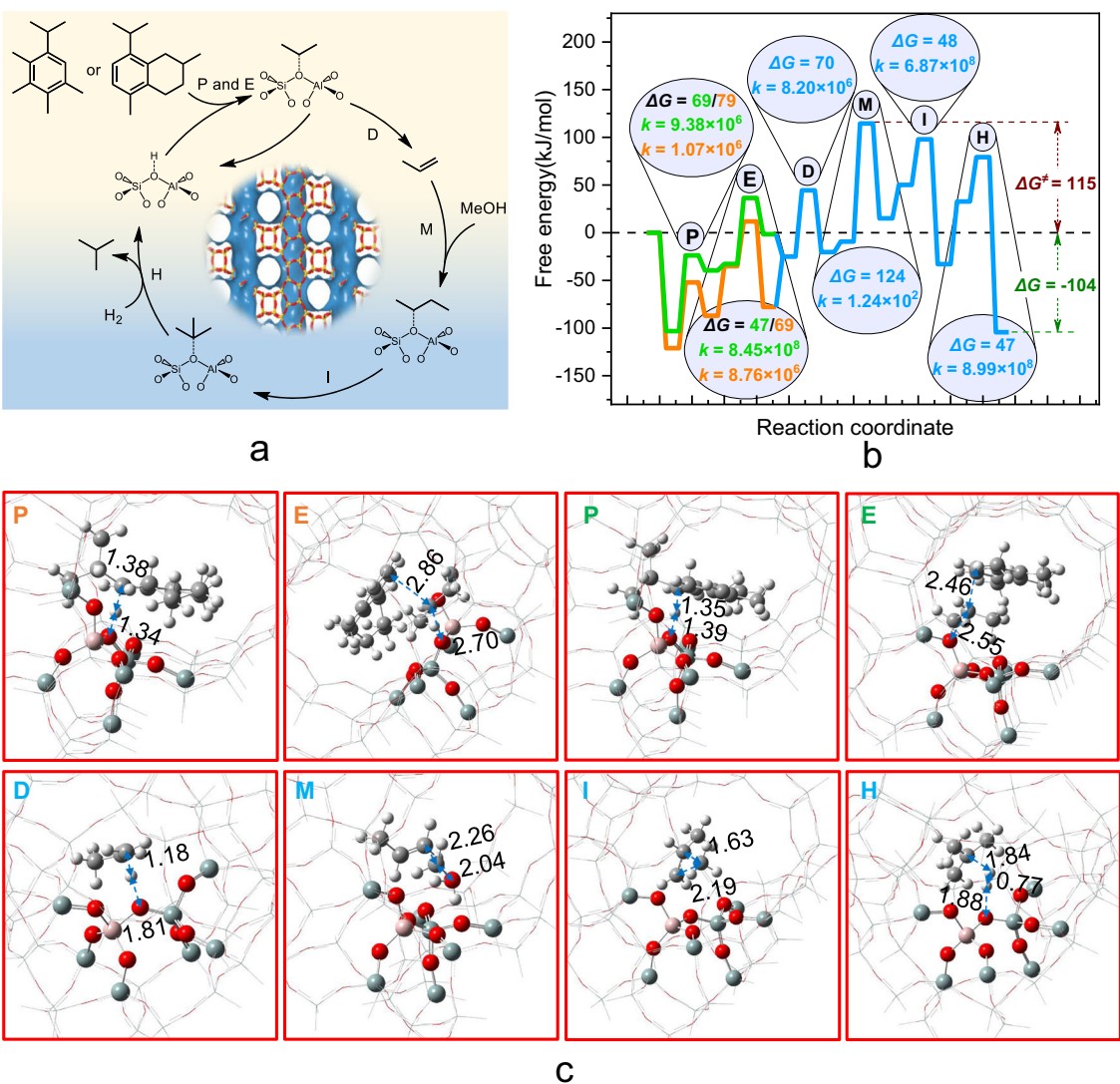

**Fig. 7 | DFT calculations. a** Reaction scheme of butane formation via the side-chain route of aromatic-based cycle. **b** Free energy profiles for butane formation from 8-isopropyl-2,5-dimethyl-1,2,3,4-tetrahydronaphthalene (orange/blue line) or 1-iso-propyl-2,3,4,5-tetramethylbenzene (green/blue line) at 315 °C over H-Beta zeolite, where the overall free energy height ($\Delta G^{\neq}$) and overall reaction free energy ($\Delta G$) in kJ mol$^{-1}$ are labeled. **c** Optimized transition states of various reaction steps, including protonation (P), elimination (E), deprotonation (D), methylation (M), isomerization (I), and hydrogenation (H); atom coloring: cyan (Si), red (O), white (H), pink (Al).

bifunctional catalyst, the indium species may facilely migrate from the oxide moiety into the zeotype moiety in the $H_2$-containing atmosphere, which can passivate the acid sites in the zeotype moiety and then rapidly deactivate the whole composite catalyst used in the $CO_2$ hydrogenation by lowering capacity of the acidic zeolite component in the transformation of methanol-related intermediates to hydrocarbons[45].

As shown in Fig. 8a, methanol and dimethyl ether (DME) are clearly detected for the $CO_2$ hydrogenation over InZrO$_x$(CP)-Beta(40) after reaction for 42 h on stream; thereafter, the selectivity to methanol and DME increases quickly, accompanied by a considerable decrease of the selectivity to butane, suggesting the rapid degeneration of the Beta zeolite. Meanwhile, more CO is generated, due to the shrinking of the methanol conversion capacity of the H-Beta zeolite that may relatively promote the competitive RWGS reaction[5]. After the reaction test, certain amounts of the In element are detected in the H-Beta moiety (Supplementary Table 4), indicating the serious phase segregation of InZrO$_x$(CP) during the $CO_2$ hydrogenation reaction. In fact, the migration of indium species turns to be a fatal defect for the application of the In-based bifunctional catalyst in practice for the

hydrogenation of $CO_2$ to hydrocarbons; it is pressing but also rather challenging to effectively inhibit the migration of the In species.

To improve the structural stability of the In-based catalyst in the $CO_2$ hydrogenation, a surface silica protection strategy was adopted in current work; that is, certain amount of $SiO_2$ (4 wt.% for InZrO$_x$(SCP-4) and 8 wt.% for InZrO$_x$(SCP-8)) was deposited on the InZrO$_x$(CP) oxide through impregnation with tetraethylorthosilicate (TEOS) solution and subsequent calcination at 500 °C (Supplementary Fig 21a). The XRD patterns shown in Fig. 9a indicate that the surface silica modification has little impact on the crystal structure of InZrO$_x$(CP). In addition, no diffraction peaks of $SiO_2$ are detected, suggesting that $SiO_2$ is highly dispersed on the InZrO$_x$ surface and/or present in amorphous phase. The EDX elemental mapping results show that the silica species are evenly distributed on the surface of InZrO$_x$(CP), despite that they cannot be clearly distinguished by XRD, HR-TEM and Aberration-corrected HAADF-STEM (Fig. 9a and Supplementary Figs. 22–24), which consolidates the high dispersion of silica in the $SiO_2$-modified InZrO$_x$ oxide. The $SiO_2$-modified InZrO$_x$(SCP-4) and InZrO$_x$(SCP-8) oxides also show larger surface area than InZrO$_x$(CP), as revealed by the $N_2$ sorption results (Supplementary Fig. 25 and

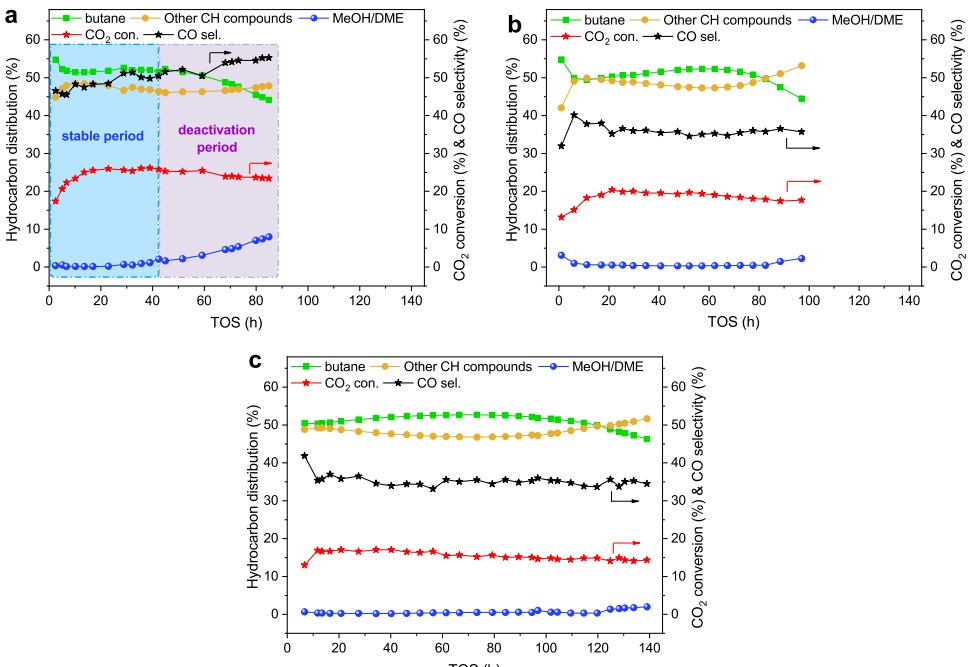

**Fig. 8 | Catalytic stability of SiO₂-modified InZrOₓ-Beta catalyst. a–c** CO₂ conversion and product distribution for CO₂ hydrogenation to butane over InZrOₓ(CP)-Beta(40) (**a**), InZrOₓ(SCP-4)-Beta(40) (**b**), and InZrOₓ(SCP-8)-Beta(40) (**c**). Reaction conditions: 315 °C, 3.0 MPa, 1200 mL g⁻¹ h⁻¹ and H₂/CO₂ = 6. The catalyst lifetime is defined as the time on stream when the selectivity to methanol and DME reaches 2% for CO₂ hydrogenation over the bifunctional catalyst.

Supplementary Table 5). Besides, the SiO₂-modified InZrOₓ(SCP-4) and InZrOₓ(SCP-8) oxides have the average particle size of 6.10 and 4.70 nm, respectively, much smaller than that of the unmodified InZrOₓ (9.76 nm) (Supplementary Fig. 26), suggesting that the silica modification can also inhibit the agglomeration of InZrOₓ upon calcination at high temperature.

Moreover, the XPS spectra shown in Fig. 9b–d illustrate that the In 3*d* and Zr 3*d* signals of silica-modified InZrOₓ(SCP-4) and InZrOₓ(SCP-8) shift towards higher binding energies, whereas the Si 2*p* signal moves to lower value, compared to the corresponding signals of the unmodified InZrOₓ(CP) counterpart. This is indicative of a strong interaction between the InZrOₓ(CP) and silica species, which is confirmed by the H₂-TPR results; the reduction of both defect In₂O₃ sites and bulk In₂O₃ in InZrOₓ(SCP-4) and InZrOₓ(SCP-8) requires higher temperature than that in the unmodified InZrOₓ(CP) counterpart (Fig. 9e). The strong interaction between the InZrOₓ and silica species is further corroborated by the calculated projected density of states (PDOS) and charge difference density (CDD) results. As shown in Supplementary Fig. 27, strong electron donation and back-donation are observed between the surface silica species and InZrOₓ oxide in the CDD plot, substantiating an intense interaction of the Si 2*p* orbitals with the In 3*d* orbitals around Fermi level. Such strong interaction makes the reduction and migration of indium species in the silica-modified InZrOₓ(SCP-4) and InZrOₓ(SCP-8) oxides more difficult than that in the unmodified InZrOₓ(CP) counterpart.

A schematic diagram is then plotted in Supplementary Fig. 21b to illustrate the mechanism of enhancing structural and catalytic stability of the InZrOₓ-Beta composite by the surface silica protection strategy. For the un-protected InZrOₓ oxide, indium species may be easily reduced to metallic In species in the reductive atmosphere, which facilely migrate to the H-Beta component and then passivate the acid sites, leading to the rapid deactivation of the composite catalyst. In contrast, after the surface silica modification, the strong interaction between the SiO₂ and InZrOₓ species can suppress the reduction of In₂O₃ to metallic indium species and then effectively hinder the metallic indium species from migration into the H-Beta component.

Consequently, the structural and catalytic stability of the InZrOₓ-Beta composite during the reduction and CO₂ hydrogenation processes can be greatly improved by the surface silica protection strategy.

The catalytic performance of InZrOₓ(SCP-4)-Beta(40) and InZrOₓ(SCP-8)-Beta(40) in the CO₂ hydrogenation was then compared with that of the InZrOₓ(CP)-Beta(40) counterpart. As shown in Fig. 8b and Supplementary Fig. 28a, the reaction time when the selectivity to unconverted methanol reaches 2% is considerably prolonged from about 42 h of InZrOₓ(CP)-Beta(40) to 97 h of InZrOₓ(SCP-4)-Beta(40) under the same conditions, indicating the higher catalytic stability of latter InZrOₓ(SCP-4)-Beta(40). Meanwhile, the acidic properties of the H-Beta zeolite component separated from various spent InZrOₓ-Beta composite catalysts after reaction for the same time were evaluated, as demonstrated in Supplementary Fig. 29 and Supplementary Table 6. Apparently, after reaction for 24 and 42 h, the total acid content and strong acid content of H-Beta zeolite separated from the spent InZrOₓ(SCP-4)-Beta(40) catalyst are both much higher than that separated from the InZrOₓ(CP)-Beta(40) counterpart. This further confirms that the surface silica protection strategy to alleviate the rapid passivation of the acid sites in H-Beta is rather effective in improving the stability of the bifunctional InZrOₓ-Beta composite catalyst in the CO₂ hydrogenation to hydrocarbons.

In addition, the selectivity to CO over InZrOₓ(SCP-4)-Beta(40) is reduced to 34.5% from the value of 51.2% over InZrOₓ(CP)-Beta(40), being much lower than those reported for the In-based bifunctional catalysts in the CO₂ hydrogenation in the literature at a similar CO₂ conversion (Supplementary Table 7). Through a further increase of the SiO₂ loading to 8 wt.%, the catalytic lifetime (e.g. the reaction time when the selectivity to unconverted methanol reaches 2%) of InZrOₓ(SCP-8)-Beta(40) is further prolonged to above 140 h, along with a lower selectivity to CO (33.1%), as shown in Fig. 8c.

Notably, although the selectivity to butane keeps at around 53% over both the SiO₂-modified and unmodified InZrOₓ-Beta catalysts during the steady stage, the selectivity to butane for the CO₂ hydrogenation over the InZrOₓ(SCP-4)-Beta(40) and InZrOₓ(SCP-8)-Beta(40) composite catalysts still decreases gradually after a long time on

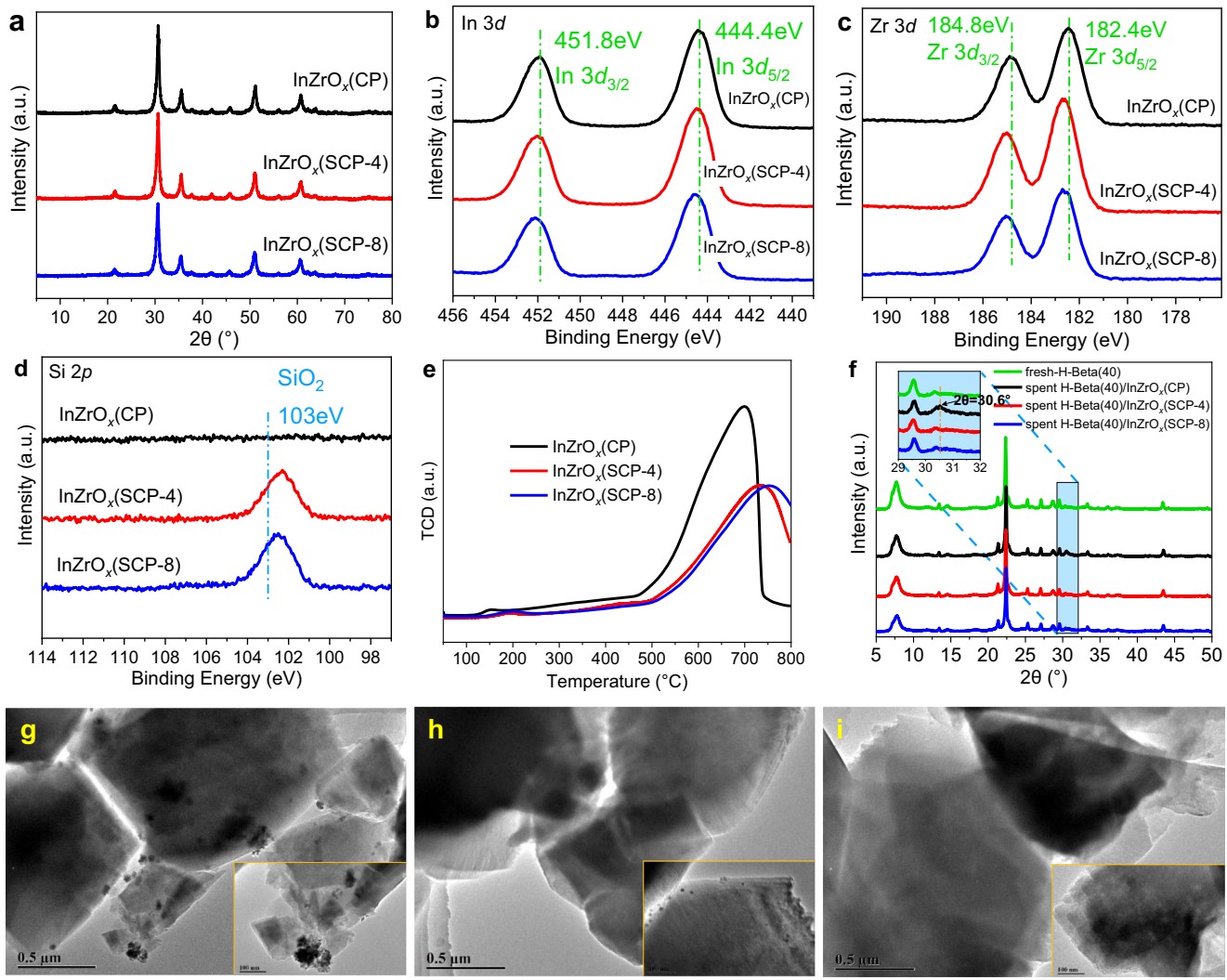

**Fig. 9 | Structure and electronic state of SiO₂-modified InZrOₓ-Beta. a** XRD patterns of fresh InZrOₓ(CP), InZrOₓ(SCP-4) and InZrOₓ(SCP-8). **b–d** In 3*d* (**b**), Zr 3*d* (**c**) and Si 2*p* (**d**) XPS spectra of fresh InZrOₓ(CP), InZrOₓ(SCP-4) and InZrOₓ(SCP-8). **e** H₂-TPR profiles of fresh InZrOₓ(CP), InZrOₓ(SCP-4) and InZrOₓ(SCP-8). **f** XRD patterns of H-Beta(40) zeolite separated from the spent InZrOₓ(CP)-Beta(40), InZrOₓ(SCP-4)-Beta(40) and InZrOₓ(SCP-8)-Beta(40) composite catalysts (The shadow insert is the enlargement of XRD patterns in the 2θ range of 29–32°). **g–i** TEM images of the spent H-Beta(40) zeolite separated from InZrOₓ(CP)-Beta(40) (**g**), InZrOₓ(SCP-4)-Beta(40) (**h**) and InZrOₓ(SCP-8)-Beta(40) (**i**) composite catalysts.

stream (ca. 70–100 h). It indicates that the surface silica modification method may not completely and eternally solve the problem of indium species migration. Nevertheless, the onset time for the decrease in the selectivity to butane is extended from 40 h of InZrOₓ(CP)-Beta(40) to ca. 70 h of InZrOₓ(SCP-4)-Beta(40) and ca. 100 h of InZrOₓ(SCP-8)-Beta(40), as demonstrated in Supplementary Fig. 28b. This also suggests that an increase of the SiO₂ loading is favorable for lowering the impact of indium migration on the butane formation. However, the deposited SiO₂ may also cover a fraction of the surface oxygen vacancies, which leads to a decrease of CO₂ adsorption capacity on the SiO₂-modified InZrOₓ oxides (Supplementary Fig. 30). As shown by the O 1s XPS spectra in Supplementary Fig. 31, InZrOₓ(SCP-4) and InZrOₓ(SCP-8) have a lower concentration of surface oxygen vacancies but abundant OH groups originated from the surface Si–OH of SiO₂, in comparison with the InZrOₓ(CP) counterpart. Consequently, the CO₂ conversion is also decreased from 25.6% over InZrOₓ(CP)-Beta(40) to 19.7% over InZrOₓ(SCP-4)-Beta(40), and further to 16.6% over InZrOₓ(SCP-8)-Beta(40). Accordingly, the loading of SiO₂ for the surface protection of the InZrOₓ oxide should be restricted to a certain value (ca. 4–8 wt.%) to elevate the catalytic stability of catalyst and meanwhile avoid a substantial decrease of the CO₂ conversion.

After the reaction test, the granules of InZrOₓ(CP) oxide and H-Beta zeolite in the InZrOₓ(CP)-Beta composite were separated from each other and then characterized by various measures. According to the TG analysis results (Supplementary Fig. 32), all three samples show a very low coking rate (ca. 0.0003–0.0005 h⁻¹), indicating that the coke deposition here should not be the major cause of catalyst deactivation. Unlike the conversion of methanol to hydrocarbons (MTH) over a zeolite catalyst in the N₂ or Ar atmosphere, for the hydrogenation of CO₂ to hydrocarbons, the presence of H₂ and H₂O in high pressure can effectively eliminate the coke precursors and thus greatly hinder the formation and accumulation of coke species. However, in the XRD patterns shown in Fig. 9f, one obvious diffraction peak at ca. 30.6° ascribed to the (222) crystal facet of In₂O₃ is distinctly detected on the H-Beta zeolite separated from the spent InZrOₓ(CP)-Beta(40) catalyst, which is further verified by the identification of InZrOₓ nano-particles in the TEM image on the surface of separated H-Beta zeolite (Fig. 9g). In contrast, such In species are nearly undetectable on the H-Beta zeolite separated from the spent InZrOₓ(SCP-4)-Beta and InZrOₓ(SCP-8)-Beta catalysts (Fig. 9h/i), suggesting that the phase segregation of InZrOₓ and the migration of indium species are effectively slowed down on the SiO₂-modified

$InZrO_x$(SCP-4) and $InZrO_x$(SCP-8) oxides during the $CO_2$ hydrogenation reaction.

In addition, after reaction for 100 h over $InZrO_x$(SCP-4) and 140 h over $InZrO_x$(SCP-8), the diffraction peaks in the XRD patterns and the binding energies in the In 3$d$, Zr 3$d$ and Si 2$p$ XPS spectra of the spent $InZrO_x$(SCP-4) and $InZrO_x$(SCP-8) oxides are highly comparable to those of the corresponding fresh ones (Supplementary Fig. 33). Meanwhile, the lattice spacing of 0.290–0.291 nm, assigned to the (222) crystal facet of $In_2O_3$, is clearly visible on the Aberration-corrected HAADF-STEM images and HR-TEM images of the spent $InZrO_x$(SCP-4) and $InZrO_x$(SCP-8) oxides (Supplementary Fig. 34), along with the uniform distribution of In, Zr, O and Si elements over these two samples (Supplementary Figs. 35, 36). The TEM images suggests that the spent $InZrO_x$(SCP-4) and $InZrO_x$(SCP-8) samples have a particle size of 7.57 and 5.16 nm, respectively, only slightly larger than the values of 6.10 and 4.70 nm for the fresh $InZrO_x$(SCP-4) and $InZrO_x$(SCP-8) counterparts, respectively (Supplementary Fig. 37). All these results reveal that the surface silica protection strategy used in current work is rather effective in suppressing the phase segregation of $InZrO_x$ oxide moiety and avoiding the rapid poisoning of the acid sites in the zeolite moiety induced by the In migration, which can thus significantly improve the structural and catalytic stability of the In-based oxide-zeolite composite catalyst in the $CO_2$ hydrogenation.

As expected, such a strategy can also be extended to the $SiO_2$-modified $In_2O_3$-Beta catalyst. As shown in Supplementary Fig. 38a, the $In_2O_3$(SCP-4)-Beta(40) catalyst shows a long catalytic lifetime (ca. 65 h) and high selectivity to butane (ca. 53% in hydrocarbons) in the $CO_2$ hydrogenation. In contrast, over the unmodified $In_2O_3$(CP)-Beta(40) counterpart, the selectivity to butane is quickly decreased to 30%, along with the generation of much more unconverted methanol (25%) after reaction for ca. 65 h (Supplementary Fig. 38b). Undoubtedly, the improved stability of the $In_2O_3$(SCP-4)-Beta(40) catalyst also originates from the inhibition of the indium species from reduction and migration by the surface silica protection, which can alleviate the rapid deactivation of the zeolite moiety in the hydrogenation of $CO_2$ to hydrocarbons (Supplementary Fig. 38c–e).

It is noteworthy that although the strong interaction between the surface $SiO_2$ species and $InZrO_x$ oxide can inhibit the indium species from easy reduction and migration and then improve the stability of the $InZrO_x$-Beta composite catalyst in the $CO_2$ hydrogenation, it does not relocate the indium species on the composite catalyst. In other words, the silica species are only highly dispersed on the surface of $InZrO_x$ oxide and do not cause any significant structural distortion and/or rearrangement of the $InZrO_x$ oxide upon the reduction and reaction process over the time. To confirm this point, the crystal structure and surface electronic states of the silica-modified $InZrO_x$(SCP-4) after reaction for different periods of time are analyzed. Apparently, the diffraction peaks in the XRD patterns, the lattice spacing in the HRTEM images and the binding energies in the In 3$d$, Zr 3$d$ and Si 2$p$ XPS spectra of the spent $InZrO_x$(SCP-4)-24h and $InZrO_x$(SCP-4)-42h samples are all highly comparable to those of the fresh counterpart (Supplementary Fig. 39j–l and Supplementary Fig. 40a). In addition, the TEM images display that the particle size of $InZrO_x$(SCP-4) is only slightly increased from 6.10 nm of fresh $InZrO_x$(SCP-4) to 6.89 nm of $InZrO_x$(SCP-4)-24h and to 7.30 nm of $InZrO_x$(SCP-4)-42h (Supplementary Fig. 39a–i). Meanwhile, the surface In/Zr and In/Si ratios of $InZrO_x$(SCP-4) also show little change upon the $CO_2$ hydrogenation reaction test (Supplementary Fig. 40b–e). That is, the major function of the introduced surface silica species is the inhibition of the indium species from reduction and migration in the reductive atmosphere containing hydrogen, whereas without causing any significant structural distortion and atomic rearrangement of the $InZrO_x$ oxide as well as the $InZrO_x$-Beta composite catalyst upon the preparation and reaction process over the time.

## Discussion

A composite bifunctional catalyst consisting of $InZrO_x$ oxide and H-Beta zeolite was designed, which exhibits excellent performance in the selective hydrogenation of $CO_2$ to butane. Under 315 °C, 3.0 MPa, and a space velocity of 1200 mL g$^{-1}$ h$^{-1}$, a high selectivity of 53.4% to butane in all hydrocarbons (CO free) is achieved at a $CO_2$ conversion of 20.4% and a selectivity of only about 2% to $CH_4$.

Various characterization measures and DFT calculation were used to explore the reaction mechanism and structure-performance relationship. The results reveal that the conversion of $CO_2$ to butane keeps to the tandem methanol-mediated mechanism and the catalytic performance of the $InZrO_x$-Beta composite is related to both the $InZrO_x$ oxide moiety and the H-Beta zeolite moiety. The generation of methanol-related intermediates by $CO_2$ hydrogenation is closely related to the surface oxygen vacancies of $InZrO_x$, which can be finely tuned through modulating the preparation methods. In contrast, the three-dimensional 12-ring channels of H-Beta zeolite conduces to forming a hydrocarbon pool (HCP) filled with higher methylbenzenes and methylnaphthalenes, which favors the successive transformation of methanol-related intermediates to butane via the aromatic-based cycle, through the alkyl side-chain elimination and subsequent methylation and hydrogenation.

In addition, to tackle the passivation of the acid sites in H-Beta by the migration of indium species of $InZrO_x$ in the reductive atmosphere containing $H_2$, a surface silica protection strategy was developed, which can effectively inhibit the phase separation of $InZrO_x$ oxide and the indium migration, and then considerably improve the catalytic stability of $InZrO_x$/Beta in the hydrogenation of $CO_2$ to hydrocarbons. The insight shown in this work may pave a way for the design of stable In-based catalyst in the $CO_2$ hydrogenation to get a specific hydrocarbon product.

## Methods
### Catalyst preparation
As described in detail in the Supplementary Information, three $InZrO_x$ oxides with an In/Zr molar ratio of 4 were prepared, viz., $InZrO_x$(CP) by co-precipitation, $InZrO_x$(SG) by sol-gel processing, and $InZrO_x$(HT) by hydrothermal method. In addition, $InZrO_x$(CP) was further modified by depositing 4 and 8 wt.% $SiO_2$ on the surface, to obtain the $SiO_2$-modified $InZrO_x$(SCP-4) and $InZrO_x$(SCP-8) oxides, respectively. Meanwhile, a series of H-Beta zeolites with a Si/Al molar ratio ($n$) of 20, 30, 40, 60, and 100 were synthesized by the hydrothermal method and denoted as Beta($n$). Moreover, H-USY (Si/Al = 5.5) and H-SSZ-13 (Si/Al = 9), purchased from Nankai University Catalyst Co., were used for comparison.

Three manners were used to compose the $InZrO_x$-Beta bifunctional catalysts, viz., dual-bed, granule-mixing, and powder-grinding. By the dual-bed manner, 0.3 g of granule $InZrO_x$ (20–40 mesh) was used as the upper layer and 0.3 g of granule H-Beta (20–40 mesh) as the lower layer. By granule-mixing, 0.3 g of $InZrO_x$ and 0.3 g of H-Beta (both in 20–40 mesh) was mixed in granules. By powder-mixing, 0.3 g of powder $InZrO_x$ and 0.3 g of powder H-Beta were grinded together for 5 min and the powdery mixture was then granulated into particles of 20–40 mesh.

### Catalyst characterization
Various catalyst samples were characterized by $N_2$ sorption, X-ray diffraction (XRD), X-ray photoelectron spectra (XPS), field emission-scanning electron microscopy (FE-SEM), transmission electron microscopy (TEM), high-resolution TEM (HRTEM), Aberration-corrected high-angle annular dark-field scanning TEM (Aberration-corrected HAADF-STEM), temperature-programmed reduction with $H_2$ ($H_2$-TPR), temperature-programmed desorption of $NH_3$ ($NH_3$-TPD) and $CO_2$ ($CO_2$-TPD), thermogravimetric analysis (TGA), inductively coupled plasma-atomic emission spectrometer (ICP-AES), in situ diffuse reflectance infrared Fourier transform (DRIFT) spectroscopy, gas chromatography-

mass spectrometry (GC-MS), and $^{12}C/^{13}C$ methanol switching experiment, as described in detail in the Supplementary Information.

## Catalytic reaction tests

The $CO_2$ hydrogenation reaction was carried out in a stainless steel tubular fixed-bed reactor, as described in detail in the Supplementary Information. Briefly, for the $CO_2$ hydrogenation to methanol, 0.3 g of $InZrO_x$ was used and the reaction was conducted under 315 °C, 3.0 MPa, and with a space velocity (SV) of 2400 mL $g^{-1} h^{-1}$ and $H_2/CO_2$ ratio of 3 in the feed. For the $CO_2$ hydrogenation to butane, unless specially claimed, 0.6 g of granule-mixed bifunctional $InZrO_x$-Beta catalyst (0.3 g $InZrO_x$ + 0.3 g H-Beta) was used and the reaction was performed under 315 °C, 3.0 MPa, a SV of 1200 mL $g^{-1} h^{-1}$, and $H_2/CO_2$ ratio of 3 in the feed. For the $InZrO_x$(CP)-Beta composite catalyst prepared by granular mixing, two components of $InZrO_x$(CP) and H-Beta in the spent $InZrO_x$(CP)-Beta catalyst after the reaction test were distinctly different in color, as displayed in Supplementary Fig. 41; therefore, two components can be easily separated by hand according to the granule color, for further characterization. For comparison, the reaction of methanol-to-olefins (MTO) was conducted in a U-type quartz tube loading with 300 mg of H-Beta and with $H_2$ as the carrier gas, under 315 °C, atmospheric pressure and a methanol weighted hourly space velocity (WHSV) of 0.05 $h^{-1}$.

## DFT calculation

DFT calculation was conducted with the Gaussian 09.E01 package, as described in detail in the Supplementary Information.

## Data availability

The source data that support the findings of this study including the article and its Supplementary Information are provided with this paper and are also available in the ScienceDB repository at https://doi.org/10.57760/sciencedb.07972 or available from the author upon reasonable request. Source data are provided with this paper.

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

## Acknowledgements

The authors sincerely thank to the financial supports of the National Key R&D Program of China (2020YFA0210900; 2018YFB0604802), National Natural Science Foundation of China (U1910203; U1862101; 21991090; 21991092; 22272195; U22A20431), Natural Science Foundation of Shanxi Province of China (202203021224009), Innovation foundation of Institute of Coal Chemistry, Chinese Academy of Sciences (SCJC-DT-2023-06), Youth Innovation Promotion Association CAS (2021172) and Excellent doctoral student award and subsidy program of Shanxi Province (BK2018001).

## Author contributions

H.W. and S.F. conducted experiments on catalyst preparation, evaluation, characterization and theoretical calculation; H.W. wrote the paper; S.G. carried out partial catalyst characterization experiments; M.D., H.Z., and W.F. provided some idea and part of the experimental guidance; S.W., Z.Q., and J.W. guided the whole experiments and revised the article; H.W. and S.F. contributed equally to this work. All the authors contributed to the discussions on the experimental and theoretical calculation results.

## Competing interests

The authors declare no competing interests.
