## [Peer Review File · Nature Communications]

Selective Conversion of CO₂ to Isobutane-enriched C₄ alkanes over InZrOx-Beta Composite CatalystREVIEWER COMMENTS

Reviewer #1 (Remarks to the Author):

Selective hydrogenation of CO₂ to value-added chemicals such as butane is very challenging, as it involves C-C coupling and modulated carbon-chain growth. In this work, the authors report that InZrOx and H-Beta composite catalysts show a promising butane selectivity of >53% along with a CO selectivity of ~35% at a CO₂ conversion of ~20%. In situ DRIFTS, isotope-labeling experiments, and DFT calculations have been combined to elucidate the reaction mechanism and pathways for the production of butane. In addition, phase separation of InZrOx oxide and transfer of indium species to zeolite are interestingly mitigated by grafting a layer of SiO₂ on its surface, which not only significantly improves catalytic stability, but also inhibits CO formation. This work is original and highly innovative, and it has made a novel and significant contribution to this emerging area. This paper is well organized and clearly written. I recommend publishing this work after minor revisions.

1. Considering that the catalytic test was performed on reduced samples, the O1s XPS spectra of reduced InZrOx oxide should be provided for estimating the real surface oxygen vacancy concentration in the reaction.
2. Please provide the coking rates of InZrOx(CP)-Beta(40), InZrOx(SCP-4)-Beta and InZrOx(SCP-8)-Beta catalysts in the reaction.
3. The authors should provide structural coordinates for optimized transition states of butane formation.

Reviewer #2 (Remarks to the Author):

This manuscript reports the combination of two catalysts, one a modified In oxide promoting CO₂ hydrogenation to methanol, and another one a zeolite transforming methanol into butane, to develop a tandem process in which CO₂ is hydrogenated selectively to a single hydrocarbon. The topic is of much interest and the results in terms of CO₂ conversion and butane selectivity are encouraging. The main novelty is the combination of the two catalysts in a granular blend, since the activity of each of the two catalyst types for each reaction is well known. Publication in Nature Communications is recommended with some changes addressing the following comments:

- The title is confusing, since it does not mention CO that is the major hydrogenation product and butane is formed in much lesser selectivity (in contrast to what the title suggests). I think that isobutane should appear somehow in the title.
- Regarding H₂-TPR, the reader is wondering why the minor peaks are commented, while the most intense peaks that can start contributing at 315 °C under constant temperature even in a larger extent are neglected.
- The influence of H₂O in the process should be addressed by adding various proportions in the feed.
- How the zeolite was separated from the InZrOx(CP)-Beta for analysis of the aromatic carbon species should be described in detail in the experimental section.

- The concept of a silica overlayer to avoid In migration to Beta zeolite is interesting, but it does not solve adequately the issue of InZrOx stability. Why does the Si overlayer increase InZrOx stability? Which is the operating mechanism?
- It is proposed that the authors use silica-coated InZrOx and determine the relocation of In on the silica layer under reaction conditions over the time on stream. Should it not the In environment change from Zr to Si over the time? Why this change does not affect to the hydrogenation performance of the catalyst? What would be the performance and stability of an InSiOx phase?
- Figure S15 and Figs. 16A and 17A should be improved to present more clearly the silica overlayer on InZrOx. Particle size distribution in Fig. S19 is difficult to be seen.
- Characterization of the exhaustively used silica-coated InZrOx by XRD and other techniques should be presented. The absence of XRD corresponding to SiO₂ does not prove its dispersion, since it could correspond to an amorphous silica phase.
- Fig. 8 does not convincingly proves catalyst stability. First the times on stream are similar for the three catalysts having different SiO₂ layer. Second, there is a decrease in the butane selectivity at times longer than 80 h, but nothing is commented on this important change.

If the authors address satisfactorily the previous points, publication would be recommended.

Response to the referees for the Manuscript NCOMMS-22-46197

To Reviewer #1:

Q0: Selective hydrogenation of CO₂ to value-added chemicals such as butane is very challenging, as it involves C-C coupling and modulated carbon-chain growth. In this work, the authors report that InZrO_x and H-Beta composite catalysts show a promising butane selectivity of >53% along with a CO selectivity of ~35% at a CO₂ conversion of ~20%. In situ DRIFTS, isotope-labeling experiments, and DFT calculations have been combined to elucidate the reaction mechanism and pathways for the production of butane. In addition, phase separation of InZrO_x oxide and transfer of indium species to zeolite are interestingly mitigated by grafting a layer of SiO₂ on its surface, which not only significantly improves catalytic stability, but also inhibits CO formation. This work is original and highly innovative, and it has made a novel and significant contribution to this emerging area. This paper is well organized and clearly written. I recommend publishing this work after minor revisions.

Response: Thanks to the reviewer for the very positive comments as well as the informative and instructive revision advices. With the help of these advices, the revised manuscript was improved greatly. We are pleased that the reviewer has clearly recognized the novelty and merits of this manuscript and hope that our revision work is satisfactory to the reviewer and the acceptance of this manuscript for publication can be approved.

Q1: Considering that the catalytic test was performed on reduced samples, the O1s XPS spectra of reduced InZrO_x oxide should be provided for estimating the real surface oxygen vacancy concentration in the reaction.

Response: Thanks to the reviewer for pointing out this important and very interesting issue.

Following the reviewer's suggestion, a series of the in situ O 1s XPS tests were conducted for various InZrO_x oxide samples holding at 400 °C in H₂ atmosphere for 2 h. The related in situ O 1s XPS spectra are displayed in Supplementary Fig. 6 and addressed in the revised manuscript (Pages 7–8), like:

“In addition, the same sequence of three oxides by the surface oxygen vacancy concentration is manifested by the in situ O 1s XPS results (Supplementary Fig. 6), viz., InZrO_x(CP) (43.47%) > InZrO_x(SG) (36.75%) > InZrO_x(HT) (31.76%). The in situ O 1s XPS gives higher surface oxygen vacancy concentrations than the ex situ one, which is ascribed to the fact that more surface oxygen defects are formed due to the elimination of certain surface

oxygen atoms by reduction in the in situ H₂-containing atmosphere, in agreement with previous works (*Chem.* 2022, 8, 1376; *J. Catal.* 2022, 413, 923).”

Supplementary Fig. 6. In situ O 1s XPS spectra of various InZrO_x oxides after holding at 400 °C in H₂ atmosphere for 2 h.

Q2: Please provide the coking rates of InZrO_x(CP)-Beta(40), InZrO_x(SCP-4)-Beta and InZrO_x(SCP-8)-Beta catalysts in the reaction.

Response: Thanks to the reviewer for the valuable suggestion, following which the coking rates of InZrO_x(CP)-Beta(40), InZrO_x(SCP-4)-Beta(40) and InZrO_x(SCP-8)-Beta(40) in the CO₂ hydrogenation were evaluated by the TG analyses in the revised manuscript. According to the TG analysis results (Supplementary Fig. 32), all three samples show a very low coking rate (ca. 0.0003–0.0005 h⁻¹), indicating that the coke deposition here should not be the major cause of catalyst deactivation. (Page 28)

Supplementary Fig. 32. TGA curves of the spent InZrO_x(CP)-Beta(40), InZrO_x(SCP-4)-Beta(40) and InZrO_x(SCP-8)-Beta(40) catalysts after the CO₂ hydrogenation reaction test.

It should be mentioned that unlike the conversion of methanol to hydrocarbons (MTH) over a zeolite catalyst in the N₂ or Ar atmosphere, for the hydrogenation of CO₂ to hydrocarbons, the presence of H₂ and H₂O in high pressure can effectively eliminate the coke precursors and thus greatly hinder the formation and accumulation of coke species. (*Nat. Catal.* 2018, 1, 666; *Nat. Catal.* 2022, 5, 1038). As a result, the coke deposition should not be the major cause of catalyst deactivation in the hydrogenation of CO₂ to hydrocarbons.

Taking the reviewer's concern into account, above points were strengthened in the revised manuscript (Page 28).

Q3: *The authors should provide structural coordinates for optimized transition states of butane formation.*

Response: Thanks to the reviewer for the valuable suggestion. Following the reviewer's suggestion, the structural coordinates for optimized transition states of butane formation have been provided in the Supplementary Data.

To Reviewer #2:

Q0: *This manuscript reports the combination of two catalysts, one a modified In oxide promoting CO₂ hydrogenation to methanol, and another one a zeolite transforming methanol into butane, to develop a tandem process in which CO₂ is hydrogenated selectively to a single hydrocarbon. The topic is of much interest and the results in terms of CO₂ conversion and butane selectivity are encouraging. The main novelty is the combination of the two catalysts in a granular blend, since the activity of each of the two catalyst types for each reaction is well known. Publication in Nature Communications is recommended with some changes addressing the following comments.*

Response: Thanks to the reviewer for the very positive comments as well as the informative and instructive revision advices. With the help of these advices, the revised manuscript was improved greatly. We are pleased that the reviewer has clearly recognized the novelty and merits of this manuscript and hope that our revision work is satisfactory to the reviewer and the acceptance of this manuscript for publication can be approved.

Q1: *The title it is confusing, since it does not mention CO that is the major hydrogenation product and butane is formed in much lesser selectivity (in contrast to what the title suggests). I think that isobutane*

should appear somehow in the title.

Response: Thanks to the reviewer for pointing out this important and very interesting issue.

Due to the intervention from the reverse water-gas shift (RWGS) reaction, the hydrogenation of CO₂ into hydrocarbons often displays a rather high selectivity to CO. An ideal solution to this issue is naturally to design a catalyst that is only active to the hydrogenation of CO₂ to the defined products but absolutely inactive to the RWGS reaction; this is however rather difficult and may be even impossible in practice.

As pointed out by the reviewer, although current InZrO_x-Beta composite catalyst shows a high selectivity to butane in hydrocarbons, a large amount of CO is simultaneously produced in the CO₂ hydrogenation process. This is also the common feature of the reported bifunctional catalysts at present for the CO₂ hydrogenation to hydrocarbons (*Nat. Commun.* 2018, 9, 3457; *Joule*. 2019, 3, 570; *ACS Catal.* 2019, 9, 3866; *Sci. Adv.* 2020, 6, eaba5433...). To avoid misleading readers, we further emphasize in the abstract, main text and conclusion that the selectivity to butane was calculated based on total hydrocarbons (CO free), while the formation of CO in the CO₂ hydrogenation process was separately evaluated, as described in previous articles (*Nat. Chem.* 2017, 9, 1019; *Angew. Chem. Int. Ed.* 2021, 60, 17735; *Nat. Catal.* 2022, 5, 1038...).

For the hydrogenation of CO₂ to hydrocarbons, the reaction results are usually reported in the product selectivity in which the product of CO from the RWGS reaction is excluded. We admit that this will make the reaction results somewhat good-looking; however, this is also the customary way to evaluate the capacity of a catalyst in the CO₂ hydrogenation to hydrocarbons, as CO may be considered as an unreacted component and can be recycled for further hydrogenation to hydrocarbons.

Meanwhile, as reported in our previous works (*Ind. Eng. Chem. Res.* 2022, 61, 17027–17038; *J. Fuel Chem. Technol.* 2023, 51, 482–491), for the hydrogenation of CO₂, although the RWGS reaction may have a severe influence on the equilibrium conversion of CO₂ as well as the equilibrium selectivity to CO, it only has a relatively minor impact on the C-based equilibrium yield of the target hydrocarbon product under appropriate reaction temperatures and pressures. Moreover, the reaction tail gas after extracting the target product, which is surely a mixture of CO and CO₂, is also credible for the production of hydrocarbons in practice through hydrogenation.

As we mentioned in a previous work (*J. Fuel Chem. Technol.* 2023, 51, 482–491), the mixture of CO and CO₂ is practical and may be even more efficient and cost-effective to produce hydrocarbons from CO₂ through hydrogenation, in comparison with pure CO₂, where the overall C-based yield should be used as the major index to evaluate such reaction processes. Naturally, this will need to devote great effort in the design and development of efficient catalysts that are highly active for the hydrogenation of both CO and CO₂ to the target hydrocarbon products rather than in the possibly hopeless obstruction of the WGS/RWGS reaction. Nevertheless, this is really far beyond the scope of current work.

In addition, iso-butane accounts for more than 86% of two butane isomers for the CO₂ hydrogenation over current InZrO_x-Beta composite catalyst (Page 12). Therefore, following the reviewer's suggestion, the Title of the revised manuscript was further modified to “*Selective Conversion of CO₂ to Isobutane-enriched C₄ alkanes over InZrO_x-Beta Composite Catalyst*”.

Thanks for the valuable suggestions. Taking the reviewer's concern into account, above points were strengthened in the revised manuscript (Pages 12–13).

Q2: *Regarding H₂-TPR, the reader is wondering why the minor peaks are commented, while the most intense peaks that can start contributing at 315 °C under constant temperature even in a larger extent are neglected.*

Response: Thanks to the reviewer for pointing out this very important issue. We did pay more attention on the minor peak in the H₂-TPR profiles, just because the intensity of such minor peak at a lower temperature is more closely related to the catalytic performance of InZrO_x-Beta composite in the CO₂ hydrogenation.

In the H₂-TPR profiles, the peak at around 150–250 °C is attributed to the reduction of defect In₂O₃ sites to In₂O_{3-x}, whereas that centered at 650 °C corresponds to the reduction of bulk In₂O₃ to metallic indium (*ACS Catal.* 2020, 10, 1133; *Angew. Chem. Int. Ed.* 2016, 55, 6261). It can be seen that the reduction peak at around 150–250 °C of InZrO_x(CP) is more intense than that of InZrO_x(SG) and InZrO_x(HT), along with certain shift towards a lower temperature. This is indicative of the presence of more surface defects or oxygen vacancies in the InZrO_x(CP) after H₂ reduction, in a good agreement with the in situ XRD Rietveld refinement (Fig. 2c in main text and Supplementary Fig. 7) and the in situ O 1s XPS deconvolution results (Supplementary Fig. 6).

As stated in the manuscript, the catalyst samples were reduced at 400 °C and used in the CO₂ hydrogenation reaction at about 315 °C. In the H₂-TPR profiles, the low-temperature reduction peak is related to the reduction of defect In₂O₃ sites to In₂O_{3-x}, which are considered to be responsible for the adsorption and activation of CO₂. In contrast, the intense peak at about 650 °C attributed to the reduction of bulk In₂O₃ to metallic indium species was less relevant to the catalytic performance of InZrO_x-Beta in the CO₂ hydrogenation.

Nevertheless, as pointed out by the reviewer, the onset temperature for the reduction of bulk In₂O₃ to metallic indium species is only around 315 °C. It is then conceivable that certain metallic indium species may be generated on the surface of InZrO_x oxide during the reduction and subsequent reaction processes. These metallic indium species may easily migrate to the zeolite moiety and then poison the acid sites of zeolite component (*Angew. Chem. Int. Ed.* 2021, 60, 17735; *J. Catal.* 2022, 413, 923). This is confirmed by observing considerable number of metallic indium particles in the H-Beta component separated from the spent InZrO_x(CP)-Beta catalyst (Fig. 9f,g in main text). This also substantiates the conclusion that the passivation of the acid sites in the H-Beta component by the indium species migrated from the InZrO_x moiety, rather than the very slow coke deposition (Supplementary Fig. 32, see the response to Q2 raised by Reviewer 1), is the main cause for the deactivation of the InZrO_x-Beta composite catalyst in the CO₂ hydrogenation.

Thanks really for the valuable comments. Considering the reviewer's concern, these relevant descriptions about the H₂-TPR profiles have been supplemented and above points were strengthened in the revised manuscript (Page 9).

Q3: *The influence of H₂O in the process should be addressed by adding various proportions in the feed.*

Response: Thanks to the reviewer for the constructive suggestion. Following this suggestion, the role of water during the CO₂ hydrogenation was further analyzed in the revised manuscript by adding different proportions of water into the H₂ and CO₂ feed (Pages 12–13; Supplementary Figs. 11–15 and Supplementary Table 3), like:

“As water is a co-product in the hydrogenation of CO₂, which acts also vividly in the competitive reverse water-gas (RWGS) reaction (*Ind. Eng. Chem. Res.* 2022, 61, 17027–17038; *J. Fuel Chem. Technol.* 2023, 51, 482–491), the possible role of water in the reaction process was further evaluated by adding different proportions of water into the H₂ and CO₂ feed. As shown in Supplementary Fig. 11a, when water is introduced into the

reaction system after about 16 h, the CO₂ conversion decreases from 19.1% to 8.9%, along with the attenuation of the selectivity to CO from 54.7% to 34.0%. Such a phenomenon becomes more evident when the proportion of water in the feed increases from 7.5% to 15.1% (Supplementary Fig. 11b), where the CO₂ conversion and selectivity to CO decreases considerably to 7.3% and 26.1%, respectively. This can be explained by the fact that more water in the reaction mixture can effectively counteract the RWGS reaction ($\text{CO}_2 + \text{H}_2 = \text{CO} + \text{H}_2\text{O}$), leading to the decline of the CO₂ conversion and the selectivity to CO (*ACS Catal.* 2020, 10, 8303; *Chem. Commun.* 2020, 56, 5239). In addition, water molecules may also compete for the active adsorption sites with the reactants, which may also abate the conversion of CO₂ to hydrocarbons.

Supplementary Fig. 11. Influence of co-feeding water on the catalytic performance of InZrO_x(CP)-Beta(40) in the CO₂ hydrogenation. Reaction conditions: 315 °C, 3.0 MPa, 1200 mL g⁻¹ h⁻¹ and H₂/CO₂ = 3. The content of co-feeding water in the feed is 7.5% (a, c) and 15.1% (b).

“It is noteworthy that the conversion of CO₂ and selectivity to CO are both spontaneously rejuvenated, when the co-feeding water is cut off (Supplementary Fig. 11c). However, the CO₂ conversion cannot be fully recovered to the original value, implying that co-feeding

water has certain impact on the catalytic activity of the InZrO_x oxide. The XRD patterns and TEM images of the spent catalysts shown in Supplementary Fig. 12 indicate that after the CO_2 hydrogenation with co-feeding water, the particle size of InZrO_x oxide increases considerably, along with the decrease of surface area and pore volume (Supplementary Fig. 13a and Supplementary Table 3). This leads to a decrease in the surface oxygen vacancies concentration, which can weaken the CO_2 adsorption capacity, as indicated by the O 1s XPS and CO_2 -TPD results (Supplementary Fig. 13b–c), although the In, Zr and O elements are still uniformly dispersed with each other in the InZrO_x oxide (Supplementary Fig. 14).

Supplementary Fig. 12. XRD patterns (a), TEM images, and the corresponding particle size distributions estimated by counting 100 NPs of the spent $\text{InZrO}_x(\text{CP})$ oxide after the CO_2 hydrogenation with co-feeding different contents of water. (b) co-feeding 7.5% water ($\text{InZrO}_x(\text{CP})$ -7.5% H_2O); (c) co-feeding 15.1% water ($\text{InZrO}_x(\text{CP})$ -15.1% H_2O).

Supplementary Fig. 13. N_2 sorption isotherms (a), O 1s XPS spectra (b), and CO_2 -TPD profiles (c) of $\text{InZrO}_x(\text{CP})$ -fresh, $\text{InZrO}_x(\text{CP})$ -7.5% H_2O , and $\text{InZrO}_x(\text{CP})$ -15.1% H_2O .

“Nevertheless, the selectivity to butane changes very little during the reaction. This is ascribed to the fact that the crystal structure, morphology, and particle size of H-Beta zeolite, which determine the manner for the formation of hydrocarbons from the methanol-related intermediates, are well maintained during the CO_2 hydrogenation with co-feeding different contents of water (Supplementary Fig. 15).

Supplementary Table 3. Textural properties of the spent $\text{InZrO}_x(\text{CP})$ oxides after conducting the CO_2 hydrogenation reaction with co-feeding different contents of water.

Sample	S_{BET} ($\text{m}^2 \text{g}^{-1}$)	S_{micro} ($\text{m}^2 \text{g}^{-1}$)	V_{micro} ($\text{cm}^3 \text{g}^{-1}$)	V_{meso} ($\text{cm}^3 \text{g}^{-1}$)
$\text{InZrO}_x(\text{CP})$ -fresh	73	29	0.006	0.31
$\text{InZrO}_x(\text{CP})$ -7.5% H_2O	39	12	0.001	0.18
$\text{InZrO}_x(\text{CP})$ -15.1% H_2O	37	12	0.001	0.17

Supplementary Fig. 14. STEM-EDX elemental mapping of the spent $\text{InZrO}_x(\text{CP})$ oxide after the CO_2 hydrogenation with co-feeding 7.5% (a) and 15.1% (b) water.

Supplementary Fig. 15. XRD patterns (a) and SEM images (b,c) of the fresh and spent H-Beta(40) zeolites after the CO_2 hydrogenation with co-feeding different contents of water. (b) co-feeding 7.5% water; (c) co-feeding 15.1% water.

Thanks really for the valuable comments. Considering the reviewer's concern, these relevant descriptions and discussion have been supplemented in the revised manuscript and above points were strengthened (Pages 12–13; Supplementary Figs. 11–15 and Supplementary Table 3).

Q4: How the zeolite was separated from the InZrO_x(CP)-Beta for analysis of the aromatic carbon species should be described in detail in the experimental section.

Response: Thanks to the reviewer for the valuable suggestion. Following this suggestion, related information has been added in the experimental section in the revised manuscript (Page 33; Supplementary Fig. 41), like:

“For the InZrO_x(CP)-Beta composite catalyst prepared by granular mixing, two components of InZrO_x(CP) and H-Beta in the spent InZrO_x(CP)-Beta catalyst after the reaction test were distinctly different in color, as displayed in Supplementary Fig. 41; therefore, two components can be easily separated by hand according to the granule color, for further characterization.”

Supplementary Fig. 41. Photos of the InZrO_x oxide and H-Beta(40) zeolite components separated from the spent InZrO_x-Beta(40) composite catalysts after the CO₂ hydrogenation reaction test.

Q5: The concept of a silica overlayer to avoid In migration to Beta zeolite is interesting, but it does not solve adequately the issue of InZrO_x stability. Why does the Si overlayer increase InZrO_x stability? Which is the operating mechanism?

Response: Thanks to the reviewer for pointing out this important and very interesting issue. In the revised manuscript, with the help of valuable hints from the reviewer’s comment, this part of work was modified greatly (Pages 21–30), like:

“The degeneration of either the metal oxide moiety or the zeotype moiety can deactivate the whole bifunctional OX-ZEO catalyst system in the hydrogenation of CO₂ to hydrocarbons. The degeneration of metal oxide often causes a rapid decrease in the CO₂ conversion, as the adsorption and activation of CO₂ are mainly performed on the surface of metal oxide. As for the acidic zeolite, it catalyzes the subsequent transformation of the methanol-related

intermediates previously generated on the oxide moiety into hydrocarbons; the rapid increase in the selectivity to unconverted methanol is an important sign for the deactivation of the zeolite component. For the In-based bifunctional catalyst, the indium species may facilely migrate from the oxide moiety into the zeotype moiety in the H₂-containing atmosphere, which can passivate the acid sites in the zeotype moiety and then rapidly deactivate the whole composite catalyst used in the CO₂ hydrogenation by lowering capacity of the acidic zeolite component in the transformation of methanol-related intermediates to hydrocarbons.” (*Angew. Chem. Int. Ed.* 2021, 60, 17735; *J. Catal.* 2022, 413, 923)

Supplementary Fig. 21. Preparation process diagram of silica modified InZrO_x(CP) (a) and schematic diagram of the function of silica modification in inhibiting the migration of indium species (b).

“To improve the structural stability of the In-based catalyst in the CO₂ hydrogenation, a surface silica protection strategy was adopted in current work; that is, certain amount of SiO₂ (4 wt.% for InZrO_x(SCP-4) and 8 wt.% for InZrO_x(SCP-8)) was deposited on the InZrO_x(CP) oxide through impregnation with tetraethylorthosilicate (TEOS) solution and subsequent calcination at 500 °C (Supplementary Fig 21a). The XRD patterns shown in Fig. 9a indicate that the surface silica modification has little impact on the crystal structure of InZrO_x(CP). In addition, no diffraction peaks of SiO₂ are detected, suggesting that SiO₂ is highly dispersed

on the InZrO_x surface and/or present in amorphous phase. The EDX elemental mapping results show that the silica species are evenly distributed on the surface of $\text{InZrO}_x(\text{CP})$, despite that they cannot be clearly distinguished by XRD, HR-TEM and Aberration-corrected HAADF-STEM (Fig. 9a and Supplementary Figs. 22–24), which consolidates the high dispersion of silica in the SiO_2 -modified InZrO_x oxide.

Fig. 9. Structure and electronic state of SiO_2 -modified InZrO_x -Beta. (a) XRD patterns of fresh $\text{InZrO}_x(\text{CP})$, $\text{InZrO}_x(\text{SCP-4})$ and $\text{InZrO}_x(\text{SCP-8})$. (b) In 3d (b), Zr 3d (c) and Si 2p (d) XPS spectra of fresh $\text{InZrO}_x(\text{CP})$, $\text{InZrO}_x(\text{SCP-4})$ and $\text{InZrO}_x(\text{SCP-8})$. (e) H_2 -TPR profiles of fresh $\text{InZrO}_x(\text{CP})$, $\text{InZrO}_x(\text{SCP-4})$ and $\text{InZrO}_x(\text{SCP-8})$. (f) XRD patterns of H-Beta(40) zeolite separated from the spent $\text{InZrO}_x(\text{CP})$ -Beta(40), $\text{InZrO}_x(\text{SCP-4})$ -Beta(40) and $\text{InZrO}_x(\text{SCP-8})$ -Beta(40) composite catalysts. TEM images of the spent H-Beta(40) zeolite separated from $\text{InZrO}_x(\text{CP})$ -Beta(40) (g), $\text{InZrO}_x(\text{SCP-4})$ -Beta(40) (h) and $\text{InZrO}_x(\text{SCP-8})$ -Beta(40) (i) composite catalysts.

“The SiO_2 -modified $\text{InZrO}_x(\text{SCP-4})$ and $\text{InZrO}_x(\text{SCP-8})$ oxides also show larger surface area than $\text{InZrO}_x(\text{CP})$, as revealed by the N_2 sorption results (Supplementary Fig. 25 and Supplementary Table 5). Besides, the SiO_2 -modified $\text{InZrO}_x(\text{SCP-4})$ and $\text{InZrO}_x(\text{SCP-8})$ oxides have the average particle size of 6.10 and 4.70 nm, respectively, much smaller than

that of the unmodified InZrO_x (9.76 nm) (Supplementary Fig. 26), suggesting that the silica modification can also inhibit the agglomeration of InZrO_x upon calcination at high temperature.

Supplementary Fig. 22. Aberration-corrected HAADF-STEM images of $\text{InZrO}_x(\text{SCP-4})$ (a,b) and $\text{InZrO}_x(\text{SCP-8})$ (d-e); HRTEM images of $\text{InZrO}_x(\text{SCP-4})$ (c) and $\text{InZrO}_x(\text{SCP-8})$ (f).

Supplementary Fig. 23. STEM and energy-dispersive X-ray (EDX) mapping images of Si (yellow), In (green), Zr (red) and O (blue) elements in $\text{InZrO}_x(\text{SCP-4})$.

Supplementary Fig. 24. STEM and energy-dispersive X-ray (EDX) mapping images of Si (yellow), In (green), Zr (red) and O (blue) elements in $\text{InZrO}_x(\text{SCP-8})$.

Supplementary Fig. 27. Projected density of states (PDOS) results of Si and In atoms and charge difference density (CDD) results of surface silica-modified $\text{InZrO}_x(\text{CP})$ oxide. The accumulation and depletion charge regions are shown in yellow and cyan, respectively, in the CDD plot.

“Moreover, the XPS spectra shown in Fig. 9b–d illustrate that the In 3d and Zr 3d signals of silica-modified $\text{InZrO}_x(\text{SCP-4})$ and $\text{InZrO}_x(\text{SCP-8})$ shift towards higher binding energies, whereas the Si 2p signal moves to lower value, compared to the corresponding signals of the unmodified $\text{InZrO}_x(\text{CP})$ counterpart. This is indicative of a strong interaction between the $\text{InZrO}_x(\text{CP})$ and silica species, which is confirmed by the H_2 -TPR results; the reduction of both defect In_2O_3 sites and bulk In_2O_3 in $\text{InZrO}_x(\text{SCP-4})$ and $\text{InZrO}_x(\text{SCP-8})$ requires higher temperature than that in the unmodified $\text{InZrO}_x(\text{CP})$ counterpart (Fig. 9e). The strong interaction between the InZrO_x and silica species is further corroborated by the calculated projected density of states (PDOS) and charge difference density (CDD) results. As shown in

Supplementary Fig. 27, strong electron donation and back-donation are observed between the surface silica species and InZrO_x oxide in the CDD plot, substantiating an intense interaction of Si 2p orbitals with In 3d orbitals around Fermi level. Such strong interaction makes the reduction and migration of indium species in the silica-modified $\text{InZrO}_x(\text{SCP-4})$ and $\text{InZrO}_x(\text{SCP-8})$ oxides more difficult than that in the unmodified $\text{InZrO}_x(\text{CP})$ counterpart.

“A schematic diagram is then plotted in Supplementary Fig. 21b to illustrate the mechanism of enhancing structural and catalytic stability of the InZrO_x -Beta composite by the surface silica protection strategy. For the un-protected InZrO_x oxide, indium species may be easily reduced to metallic In species in the reductive atmosphere, which facilely migrate to the H-Beta component and then passivate the acid sites, leading to the rapid deactivation of the composite catalyst. In contrast, after the surface silica modification, the strong interaction between the SiO_2 and InZrO_x species can suppress the reduction of In_2O_3 to metallic indium species and then effectively hinder the metallic indium species from migration into the H-Beta component. Consequently, the structural and catalytic stability of the InZrO_x -Beta composite during the reduction and CO_2 hydrogenation processes can be greatly improved by the surface silica protection strategy.

Thanks really to the reviewer for the valuable comments. Considering the reviewer’s concern, these relevant descriptions and discussion have been supplemented and above points were strengthened in the revised manuscript (Pages 21–30; Supplementary Figs. 21–27).

Q6: It is proposed that the authors use silica-coated InZrO_x and determine the relocation of In on the silica layer under reaction conditions over the time on stream. Should it not the In environment change from Zr to Si over the time? Why this change does not affect to the hydrogenation performance of the catalyst? What would be the performance and stability of an InSiO_x phase?

Response: Thanks to the reviewer for pointing out this very important and interesting issue. We are very sorry for the misunderstanding caused to the reviewer due to the unclear description of the surface silica protection mechanism in the old version manuscript. As responded to the Query Q5, with the help of these valuable hints from the reviewer’s constructive comments, this part of work was modified greatly in the revised manuscript (Pages 21–30), like:

“It is noteworthy that although the strong interaction between the surface SiO_2 species and

InZrO_x oxide can inhibit the indium species from easy reduction and migration and then improve the stability of the InZrO_x-Beta composite catalyst in the CO₂ hydrogenation, it does not relocate the indium species on the composite catalyst. In other words, the silica species are only highly dispersed on the surface of InZrO_x oxide and do not cause any significant structural distortion and/or rearrangement of the InZrO_x oxide upon the reduction and reaction process over the time.

Supplementary Fig. 39. TEM images and the corresponding particle size distributions estimated by counting 100 NPs of the spent InZrO_x(SCP-4) oxide after reaction for different times: fresh (a–c), 24 h (d–f) and 42 h (g–i); HRTEM image of the spent InZrO_x(SCP-4) sample after reaction for different times: fresh (j), 24 h (k) and 42 h (l).

“To confirm this point, the crystal structure and surface electronic states of the silica-modified InZrO_x(SCP-4) after reaction for different times are analyzed. Apparently, the diffraction peaks in the XRD patterns, the lattice spacing in the HRTEM images and the binding energies in the In 3d, Zr 3d and Si 2p XPS spectra of the spent InZrO_x(SCP-4)-24h

and $\text{InZrO}_x(\text{SCP-4})$ -42h samples are all highly comparable to those of the fresh counterpart (Supplementary Fig. 39j–l and Supplementary Fig. 40a). In addition, the TEM images display that the particle size of $\text{InZrO}_x(\text{SCP-4})$ is only slightly increased from 6.10 nm of fresh $\text{InZrO}_x(\text{SCP-4})$ to 6.89 nm of $\text{InZrO}_x(\text{SCP-4})$ -24h and to 7.30 nm of $\text{InZrO}_x(\text{SCP-4})$ -42h (Supplementary Fig. 39a–i). Meanwhile, the surface In/Zr and In/Si ratios of $\text{InZrO}_x(\text{SCP-4})$ also show little change upon the CO_2 hydrogenation reaction test (Supplementary Fig. 40b–e). That is, the major function of the introduced surface silica species is the inhibition of the indium species from reduction and migration in the reductive atmosphere containing hydrogen, whereas without causing any significant structural distortion and atomic rearrangement of the InZrO_x oxide as well as the InZrO_x -Beta composite catalyst upon the preparation and reaction process over the time.

Supplementary Fig. 40. XRD patterns (a) of InZrO_x separated from the spent $\text{InZrO}_x(\text{SCP-4})$ -Beta(40) composite catalysts after reaction for different times; Surface In/Zr and In/Si ratios of the spent $\text{InZrO}_x(\text{SCP-4})$ oxide after reaction for different times (b). The surface In/Zr and In/Si ratios were calculated by the following equations: $\text{In/Zr} = (I_{\text{In}}/S_{\text{In}})/(I_{\text{Zr}}/S_{\text{Zr}})$, $\text{In/Si} = (I_{\text{In}}/S_{\text{In}})/(I_{\text{Si}}/S_{\text{Si}})$, where I_E represent peak area of various elements (E) in the XPS spectra and S_E is the corresponding sensitivity factor. The In 3d (c), Zr 3d (d) and Si 2p (e) XPS spectra of $\text{InZrO}_x(\text{SCP-4})$ separated from the spent $\text{InZrO}_x(\text{SCP-4})$ -Beta(40) composite catalyst after reaction for different times.

“The catalytic performance of $\text{InZrO}_x(\text{SCP-4})$ -Beta(40) and $\text{InZrO}_x(\text{SCP-8})$ -Beta(40) in the CO_2 hydrogenation was then compared with that of the $\text{InZrO}_x(\text{CP})$ -Beta(40) counterpart. As

shown in Fig. 8b and Supplementary Fig. 28a, the reaction time when the selectivity to unconverted methanol reaches 2% is considerably prolonged from about 42 h of $\text{InZrO}_x(\text{CP})\text{-Beta}(40)$ to 97 h of $\text{InZrO}_x(\text{SCP-4})\text{-Beta}(40)$ under the same conditions, indicating the higher catalytic stability of latter $\text{InZrO}_x(\text{SCP-4})\text{-Beta}(40)$. Meanwhile, the acidic properties of the H-Beta zeolite component separated from various spent $\text{InZrO}_x\text{-Beta}$ composite catalysts after reaction for the same time were evaluated, as demonstrated in Supplementary Fig. 29 and Supplementary Table 6. Apparently, after reaction for 24 and 42 h, the total acid content and strong acid content of H-Beta zeolite separated from the spent $\text{InZrO}_x(\text{SCP-4})\text{-Beta}(40)$ catalyst are both much higher than that separated from the $\text{InZrO}_x(\text{CP})\text{-Beta}(40)$ counterpart. This further confirms that the surface silica protection strategy to alleviate the rapid passivation of the acid sites in H-Beta is rather effective in improving the stability of the bifunctional $\text{InZrO}_x\text{-Beta}$ composite catalyst in the CO_2 hydrogenation to hydrocarbons.

In addition, the selectivity to CO over $\text{InZrO}_x(\text{SCP-4})\text{-Beta}(40)$ is reduced to 34.5% from the value of 51.2% over $\text{InZrO}_x(\text{CP})\text{-Beta}(40)$, being much lower than those reported for the In-based bifunctional catalysts in the CO_2 hydrogenation in the literature at a similar CO_2 conversion (Supplementary Table 7). Through a further increase of the SiO_2 loading to 8 wt.%, the catalytic lifetime (e.g. the reaction time when the selectivity to unconverted methanol reaches 2%) of $\text{InZrO}_x(\text{SCP-8})\text{-Beta}(40)$ is further prolonged to above 140 h, along with a lower selectivity to CO (33.1%), as shown in Fig. 8c.

“Notably, although the selectivity to butane keeps at around 53% over both the SiO_2 -modified and unmodified $\text{InZrO}_x\text{-Beta}$ catalysts during the steady stage, the selectivity to butane for the CO_2 hydrogenation over the $\text{InZrO}_x(\text{SCP-4})\text{-Beta}(40)$ and $\text{InZrO}_x(\text{SCP-8})\text{-Beta}(40)$ composite catalysts still decreases gradually after a long time on stream (ca. 70–100 h). It indicates that the surface silica modification method may not completely and eternally solve the problem of indium species migration. Nevertheless, the onset time for the decrease in the selectivity to butane is extended from 40 h of $\text{InZrO}_x(\text{CP})\text{-Beta}(40)$ to ca. 70 h of $\text{InZrO}_x(\text{SCP-4})\text{-Beta}(40)$ and ca. 100 h of $\text{InZrO}_x(\text{SCP-8})\text{-Beta}(40)$, as demonstrated in Supplementary Fig. 28b. This also suggests that an increase of the SiO_2 loading is favorable for lowering the impact of indium migration on butane formation. However, the deposited SiO_2 may also cover a fraction of the surface oxygen vacancies, which leads to a decrease of CO_2 adsorption capacity on the SiO_2 -modified InZrO_x oxides (Supplementary Fig. 30). As shown by the O 1s XPS spectra in

Supplementary Fig. 31, $\text{InZrO}_x(\text{SCP-4})$ and $\text{InZrO}_x(\text{SCP-8})$ have a lower concentration of surface oxygen vacancies but abundant OH groups originated from the surface Si–OH of SiO_2 , in comparison with the $\text{InZrO}_x(\text{CP})$ counterpart. Consequently, the CO_2 conversion is also decreased from 25.6% over $\text{InZrO}_x(\text{CP})\text{-Beta}(40)$ to 19.7% over $\text{InZrO}_x(\text{SCP-4})\text{-Beta}(40)$, and further to 16.6% over $\text{InZrO}_x(\text{SCP-8})\text{-Beta}(40)$. Accordingly, the loading of SiO_2 for the surface protection of the InZrO_x oxide should be restricted to a certain value (ca. 4–8 wt.%) to elevate the catalytic stability of catalyst and meanwhile avoid a substantial decrease of the CO_2 conversion.

Supplementary Fig. 38. CO_2 conversion and product distribution for the CO_2 hydrogenation to butane over $\text{In}_2\text{O}_3(\text{SCP-4})\text{-Beta}(40)$ (a); Selectivity to butane and $\text{CH}_3\text{OH/DME}$ over $\text{In}_2\text{O}_3(\text{CP})\text{-Beta}(40)$ and $\text{In}_2\text{O}_3(\text{SCP-4})\text{-Beta}(40)$ (b). Reaction conditions: 315 °C, 3.0 MPa, 1200 $\text{mL g}^{-1} \text{h}^{-1}$ and $\text{H}_2/\text{CO}_2 = 6$. The catalyst lifetime is defined as the time on stream when the selectivity to unconverted methanol and DME reaches 2% for the CO_2 hydrogenation over the bifunctional catalyst; In 3d (c) and Si 2p (d) XPS spectra of fresh $\text{In}_2\text{O}_3(\text{CP})$ and $\text{In}_2\text{O}_3(\text{SCP-4})$; H_2 -TPR profiles (e) of fresh $\text{In}_2\text{O}_3(\text{CP})$ and $\text{In}_2\text{O}_3(\text{SCP-4})$.

Moreover, following the reviewer's suggestion, such a strategy can also be extended to the SiO_2 -modified $\text{In}_2\text{O}_3\text{-Beta}$ catalyst. As shown in Supplementary Fig. 38a, the $\text{In}_2\text{O}_3(\text{SCP-4})\text{-Beta}(40)$ catalyst shows a long catalytic lifetime (ca. 65 h) and high selectivity to butane (ca. 53% in hydrocarbons) in the CO_2 hydrogenation. In contrast, over the unmodified $\text{In}_2\text{O}_3(\text{CP})\text{-Beta}(40)$ counterpart, the selectivity to butane is quickly decreased to 30%, along with the generation of much more unconverted methanol (25%) after reaction for

ca. 65 h (Supplementary Fig. 38b). Undoubtedly, the improved stability of the $\text{In}_2\text{O}_3(\text{SCP-4})\text{-Beta}(40)$ catalyst also originates from the inhibition of the indium species from reduction and migration by the surface silica protection, which can alleviate the rapid deactivation of the zeolite moiety in the hydrogenation of CO_2 to hydrocarbons (Supplementary Fig. 38c–e).

Thanks again to the reviewer for the valuable comments. Considering the reviewer's concern, these relevant descriptions and discussion have been supplemented and above points were strengthened in the revised manuscript (Pages 21–30).

Q7: Figure S15 and Figs. 16A and 17A should be improved to present more clearly the silica overlayer on InZrO_x . Particle size distribution in Fig. S19 is difficult to be seen.

Response: Thanks really for raising these matters present in the old version manuscript.

Supplementary Fig. 22. Aberration-corrected HAADF-STEM images of $\text{InZrO}_x(\text{SCP-4})$ (a,b) and $\text{InZrO}_x(\text{SCP-8})$ (d–e); HRTEM images of $\text{InZrO}_x(\text{SCP-4})$ (c) and $\text{InZrO}_x(\text{SCP-8})$ (f).

Considering the reviewer's suggestion, the morphology and structure of silica-modified $\text{InZrO}_x(\text{SCP-4})$ and $\text{InZrO}_x(\text{SCP-8})$ were further investigated by Aberration-corrected HAADF-STEM. As shown in Supplementary Fig. 22, no evident silica overlayer is observed even in the high resolution Aberration-corrected HAADF-STEM images, indicating that

these silica species are amorphous and evenly dispersed on the surface of $\text{InZrO}_x(\text{SCP-4})$ and $\text{InZrO}_x(\text{SCP-8})$.

This is also supported by the EDX elemental mapping that Si, In, Zr and O elements are uniformly dispersed with each other over these two samples (Supplementary Figs. 23 and 24). Moreover, the HR-TEM images show that the lattice spacing of 0.290 nm for the (222) crystal facet of In_2O_3 extend to the edge of catalyst, whereas the nonmetallic layer (e.g. silica overlayer) is undetectable (Supplementary Fig. 22).

Supplementary Fig. 23. STEM and energy-dispersive X-ray (EDX) mapping of Si (yellow), In (green), Zr (red) and O (blue) elements in $\text{InZrO}_x(\text{SCP-4})$.

Supplementary Fig. 24. STEM and energy-dispersive X-ray (EDX) mapping of Si (yellow), In (green), Zr (red) and O (blue) elements in $\text{InZrO}_x(\text{SCP-8})$.

Moreover, according to the reviewer's suggestion, the TEM images and corresponding

particle size distributions of the $\text{InZrO}_x(\text{SCP-4})$ and $\text{InZrO}_x(\text{SCP-8})$ samples were further improved to make them clearer (Supplementary Fig. 26).

Supplementary Fig. 26. TEM images and corresponding particle size distributions estimated by counting 100 NPs of $\text{InZrO}_x(\text{SCP-4})$ (a–c) and $\text{InZrO}_x(\text{SCP-8})$ (d–f).

It is noteworthy that the silica layer deposited on InZrO_x may also cover a fraction of the surface oxygen vacancies and then has certain impact on the adsorption and activation of H_2 and CO_2 on the catalyst surface. Although the increase of the SiO_2 loading is favorable for lowering the impact of indium migration on the butane formation, it also leads to a decrease of CO_2 adsorption capacity on the SiO_2 -modified InZrO_x oxides (Supplementary Fig. 30). As shown by the O 1s XPS spectra in Supplementary Fig. 31, $\text{InZrO}_x(\text{SCP-4})$ and $\text{InZrO}_x(\text{SCP-8})$ have a lower concentration of surface oxygen vacancies but abundant OH groups originated from the surface Si–OH of SiO_2 , in comparison with the $\text{InZrO}_x(\text{CP})$ counterpart. Consequently, the CO_2 conversion is also decreased from 25.6% over $\text{InZrO}_x(\text{CP})\text{-Beta}(40)$ to 19.7% over $\text{InZrO}_x(\text{SCP-4})\text{-Beta}(40)$, and further to 16.6% over $\text{InZrO}_x(\text{SCP-8})\text{-Beta}(40)$. Accordingly, the loading of SiO_2 for the surface protection of the InZrO_x oxide should be restricted to a certain value (ca. 4–8 wt.%) to elevate the catalytic stability of catalyst and meanwhile avoid a substantial decrease of the CO_2 conversion. (Pages 27–28)

Thanks again to the reviewer for the valuable suggestions. Considering the reviewer’s concern, all of these descriptions and improvement have been embodied in the revised manuscript and above points were strengthened in the revised manuscript (Pages 21–30).

Q8: Characterization of the exhaustively used silica-coated InZrO_x by XRD and other techniques should be presented. The absence of XRD corresponding to SiO_2 does not prove its dispersion, since it could

correspond to an amorphous silica phase.

Response: Thanks a lot to the reviewer for the timely comments. Considering the reviewer's concerns, the crystal structure, particle size, surface electronic states and elemental distribution of exhaustively used silica-modified $\text{InZrO}_x(\text{SCP-4})$ and $\text{InZrO}_x(\text{SCP-8})$ catalysts after the CO_2 hydrogenation reaction tests were carefully investigated in the revised manuscript (Pages 28–29), like:

Supplementary Fig. 33. XRD patterns (a) and In 3d (c), Zr 3d (d) and Si 2p (e) XPS spectra of the fresh and spent $\text{InZrO}_x(\text{SCP-4})$ oxides after carrying out the CO_2 hydrogenation for 100 h; XRD patterns (b) and In 3d (f), Zr 3d (g) and Si 2p (h) XPS spectra of the fresh and spent $\text{InZrO}_x(\text{SCP-8})$ oxides after conducting the CO_2 hydrogenation reaction for 140 h.

“In addition, after reaction for 100 h over $\text{InZrO}_x(\text{SCP-4})$ and 140 h over $\text{InZrO}_x(\text{SCP-8})$, the diffraction peaks in the XRD patterns and the binding energies in the In 3d, Zr 3d and Si 2p XPS spectra of the spent $\text{InZrO}_x(\text{SCP-4})$ and $\text{InZrO}_x(\text{SCP-8})$ oxides are highly comparable to those of the corresponding fresh ones (Supplementary Fig. 33). Meanwhile, the lattice

spacing of 0.290–0.291 nm, assigned to the (222) crystal facet of In_2O_3 , is clearly visible on the Aberration-corrected HAADF-STEM images and HR-TEM images of the spent $\text{InZrO}_x(\text{SCP-4})$ and $\text{InZrO}_x(\text{SCP-8})$ oxides (Supplementary Fig. 34), along with the uniform distribution of In, Zr, O and Si elements over these two samples (Supplementary Figs. 35 and 36). The TEM images suggests that the spent $\text{InZrO}_x(\text{SCP-4})$ and $\text{InZrO}_x(\text{SCP-8})$ samples have a particle size of 7.57 and 5.16 nm, respectively, only slightly larger than the values of 6.10 and 4.70 nm for the fresh $\text{InZrO}_x(\text{SCP-4})$ and $\text{InZrO}_x(\text{SCP-8})$ counterparts, respectively (Supplementary Fig. 37).

Supplementary Fig. 34. Aberration-corrected HAADF-STEM and HR-TEM images of the spent $\text{InZrO}_x(\text{SCP-4})$ oxide after conducting the CO_2 hydrogenation reaction for 100 h (a,b) and the spent $\text{InZrO}_x(\text{SCP-8})$ oxide after conducting the CO_2 hydrogenation reaction for 140 h (c,d).

Supplementary Fig. 35. STEM and energy-dispersive X-ray (EDX) elemental mapping of Si (yellow), In (green), Zr (red) and O (blue) of the spent $\text{InZrO}_x(\text{SCP-4})$ oxide after carrying out the CO_2 hydrogenation reaction for 100 h.

All these results reveal that the surface silica protection strategy used in current work is rather effective in suppressing the phase segregation of InZrO_x oxide moiety and avoiding

the rapid poisoning of the acid sites in the zeolite moiety induced by the In migration, which can thus significantly improve the structural and catalytic stability of the In-based oxide-zeolite composite catalyst in the CO₂ hydrogenation.

Supplementary Fig. 36. STEM and energy-dispersive X-ray (EDX) elemental mapping of Si (yellow), In (green), Zr (red) and O (blue) of the spent InZrO_x(SCP-8) oxide after carrying out the CO₂ hydrogenation reaction for 140 h.

Supplementary Fig. 37. TEM images and the corresponding particle size distributions estimated by counting 100 NPs of the spent InZrO_x(SCP-4) oxide after conducting the CO₂ hydrogenation reaction for 100 h (a–c) and the spent InZrO_x(SCP-8) oxide after conducting the CO₂ hydrogenation reaction for 140 h (d–f).

We do agree with the reviewer’s opinion that the absence of XRD diffraction peaks of SiO₂ species does not fully prove its dispersion, as these SiO₂ species may be amorphous phase.

Therefore, the Aberration-corrected HAADF-STEM and corresponding EDX elemental mapping were further carried out; the results indicate that these amorphous silica species are evenly dispersed on the surface of InZrO_x oxide (Supplementary Figs. 22–24). Accordingly, these relevant descriptions have been supplemented in the revised manuscript (Pages 23–24) like:

Supplementary Fig. 22. Aberration-corrected HAADF-STEM images of $\text{InZrO}_x(\text{SCP-4})$ (a,b) and $\text{InZrO}_x(\text{SCP-8})$ (d–e); HRTEM images of $\text{InZrO}_x(\text{SCP-4})$ (c) and $\text{InZrO}_x(\text{SCP-8})$ (f).

“The XRD patterns shown in Fig. 9a indicate that the surface silica modification has little impact on the crystal structure of $\text{InZrO}_x(\text{CP})$. In addition, no diffraction peaks of SiO_2 are detected, suggesting that SiO_2 is highly dispersed on the InZrO_x surface and/or present in amorphous phase. The EDX elemental mapping results show that the silica species are evenly distributed on the surface of $\text{InZrO}_x(\text{CP})$, despite that they cannot be clearly distinguished by XRD, HR-TEM and Aberration-corrected HAADF-STEM (Fig. 9a and Supplementary Figs. 22–24), which consolidates the high dispersion of silica in the SiO_2 -modified InZrO_x oxide. The SiO_2 -modified $\text{InZrO}_x(\text{SCP-4})$ and $\text{InZrO}_x(\text{SCP-8})$ oxides also show larger surface area than $\text{InZrO}_x(\text{CP})$, as revealed by the N_2 sorption results (Supplementary Fig. 25 and Supplementary Table 5). Besides, the SiO_2 -modified $\text{InZrO}_x(\text{SCP-4})$ and $\text{InZrO}_x(\text{SCP-8})$ oxides have the average particle size of 6.10 and 4.70 nm, respectively, much smaller than that of the unmodified InZrO_x (9.76 nm) (Supplementary Fig.

26), suggesting that the silica modification can also inhibit the agglomeration of InZrO_x upon calcination at high temperature.

Supplementary Fig. 23. STEM and energy-dispersive X-ray (EDX) mapping of Si (yellow), In (green), Zr (red) and O (blue) elements in $\text{InZrO}_x(\text{SCP-4})$.

Supplementary Fig. 24. STEM and energy-dispersive X-ray (EDX) mapping of Si (yellow), In (green), Zr (red) and O (blue) elements in $\text{InZrO}_x(\text{SCP-8})$.

Thanks again to the reviewer for the valuable suggestions. Considering the reviewer's concern, all of these descriptions and improvement have been embodied in the revised manuscript and above points were strengthened in the revised manuscript (Pages 21–30).

Q9: Fig. 8 does not convincingly prove catalyst stability. First the times on stream are similar for the three catalysts having different SiO_2 layer. Second, there is a decrease in the butane selectivity at times longer than 80 h, but nothing is commented on this important change.

Response: Thanks to the reviewer for pointing out this important and very interesting issue. We are sorry for not providing a clear definition and clarification in the old manuscript on the stability of the bifunctional InZrO_x-Beta composite catalyst in the CO₂ hydrogenation to hydrocarbons.

As responded to previous Query Q5, the degeneration of either the metal oxide moiety or the zeotype moiety can deactivate the whole bifunctional OX-ZEO catalyst system in the hydrogenation of CO₂ to hydrocarbons (*Chem. Soc. Rev.* 2019, 48, 3193; *ACS Catal.* 2019, 9, 3026). The degeneration of metal oxide often causes a rapid decrease in the CO₂ conversion, as the adsorption and activation of CO₂ are mainly performed on the surface of metal oxide (*Nat. Chem.* 2017, 9, 1019; *ACS Catal.* 2020, 10, 1133). As for the acidic zeolite, it catalyzes the transformation of the methanol-related intermediates previously generated on the oxide moiety into hydrocarbons; the rapid increase in the selectivity to unconverted methanol is an important sign for the deactivation of the zeolite component (*Nat. Commun.* 2019,10, 1297; *Angew. Chem. Int. Ed.* 2021, 60, 17735; *ACS Catal.* 2021, 11, 9729). For the In-based bifunctional catalyst, the indium species may facilely migrate from the oxide moiety into the zeotype moiety in the H₂-containing atmosphere, which can passivate the acid sites in the zeotype moiety and then rapidly deactivate the whole composite catalyst used in the CO₂ hydrogenation by lowering capacity of the acidic zeolite component in the transformation of methanol-related intermediates to hydrocarbons.” (*Angew. Chem. Int. Ed.* 2021, 60, 17735; *J. Catal.* 2022, 413, 923)

In this work, the gradual increase of the selectivity to unconverted methanol reveals that the deactivation of InZrO_x-Beta bifunctional catalyst is mainly ascribed to the passivation of the acid sites in the H-Beta zeolite, although the CO₂ conversion is relative stable with the reaction time. Herein, the catalytic lifetime of the bifunctional InZrO_x-Beta composite catalyst was defined as the reaction time when the selectivity to unconverted methanol was increased to 2%.

As shown in Fig. 8b and Supplementary Fig. 28a, the reaction time when the selectivity to unconverted methanol reaches 2% is considerably prolonged from about 42 h of InZrO_x(CP)-Beta(40) to 97 h of InZrO_x(SCP-4)-Beta(40) and 140 h of InZrO_x(SCP-8)-Beta(40) under the same conditions, indicating the higher catalytic stability of latter InZrO_x(SCP-4)-Beta(40) and InZrO_x(SCP-8)-Beta(40).

Supplementary Fig. 28. Selectivity to CH₃OH/DME (a) and butane (b) over the InZrO_x(CP)-Beta(40), InZrO_x(SCP-4)-Beta(40) and InZrO_x(SCP-8)-Beta(40) composite catalysts. Reaction conditions: 315 °C, 3.0 MPa, 1200 mL g⁻¹ h⁻¹, and H₂/CO₂ = 6.

Meanwhile, the acidic properties of the H-Beta zeolite component separated from various spent InZrO_x-Beta composite catalysts after reaction for the same time were evaluated, as demonstrated in Supplementary Fig. 29 and Supplementary Table 6. Apparently, after reaction for 24 and 42 h, the total acid content and strong acid content of H-Beta zeolite separated from the spent InZrO_x(SCP-4)-Beta(40) catalyst are both much higher than that separated from the InZrO_x(CP)-Beta(40) counterpart. This further confirms that the surface silica protection strategy to alleviate the rapid passivation of the acid sites in H-Beta is rather effective in improving the stability of the bifunctional InZrO_x-Beta composite catalyst in the CO₂ hydrogenation to hydrocarbons.

Supplementary Fig. 29. NH₃-TPD profiles of the H-Beta(40) zeolite separated from the spent InZrO_x(CP)-Beta(40) and InZrO_x(SCP-4)-Beta(40) composite catalysts after carrying out the CO₂ hydrogenation reaction for 24 h and 42 h.

Supplementary Table 6. Acidic content of various H-Beta(40) zeolite separated from the spent $\text{InZrO}_x(\text{CP})\text{-Beta}(40)$ and $\text{InZrO}_x(\text{SCP-4})\text{-Beta}(40)$ composite catalysts after conducting the CO_2 hydrogenation reaction for 24 and 42 h.

H-Beta(40) zeolite separated from	Acid content ($\mu\text{mol g}^{-1}$)	
	weak	strong
H-Beta(40)/ $\text{InZrO}_x(\text{CP})$, reaction for 24 h	127	101
H-Beta(40)/ $\text{InZrO}_x(\text{CP})$, reaction for 42 h	108	86
H-Beta(40)/ $\text{InZrO}_x(\text{SCP-4})$, reaction for 24 h	230	189
H-Beta(40)/ $\text{InZrO}_x(\text{SCP-4})$, reaction for 42 h	169	133

Indeed, as the reviewer pointed out, although the selectivity to butane keeps at around 53% over both the SiO_2 -modified and unmodified $\text{InZrO}_x\text{-Beta}$ catalysts during the steady stage, the selectivity to butane for the CO_2 hydrogenation over the $\text{InZrO}_x(\text{SCP-4})\text{-Beta}(40)$ and $\text{InZrO}_x(\text{SCP-8})\text{-Beta}(40)$ composite catalysts still decreases gradually after a long time on stream (ca. 70–100 h). It indicates that the surface silica modification method may not completely and eternally solve the problem of indium species migration. Nevertheless, the onset time for the decrease in the selectivity to butane is extended from 40 h of $\text{InZrO}_x(\text{CP})\text{-Beta}(40)$ to ca. 70 h of $\text{InZrO}_x(\text{SCP-4})\text{-Beta}(40)$ and ca. 100 h of $\text{InZrO}_x(\text{SCP-8})\text{-Beta}(40)$, as demonstrated in Supplementary Fig. 28b. This also suggests that an increase of the SiO_2 loading is favorable for lowering the impact of indium migration on the butane formation. However, the deposited SiO_2 may also cover a fraction of the surface oxygen vacancies, which leads to a decrease of CO_2 adsorption capacity on the SiO_2 -modified InZrO_x oxides (Supplementary Fig. 30). As shown by the O 1s XPS spectra in Supplementary Fig. 31, $\text{InZrO}_x(\text{SCP-4})$ and $\text{InZrO}_x(\text{SCP-8})$ have a lower concentration of surface oxygen vacancies but abundant OH groups originated from the surface Si–OH of SiO_2 , in comparison with the $\text{InZrO}_x(\text{CP})$ counterpart. Consequently, the CO_2 conversion is also decreased from 25.6% over $\text{InZrO}_x(\text{CP})\text{-Beta}(40)$ to 19.7% over $\text{InZrO}_x(\text{SCP-4})\text{-Beta}(40)$, and further to 16.6% over $\text{InZrO}_x(\text{SCP-8})\text{-Beta}(40)$. Accordingly, the loading of SiO_2 for the surface protection of the InZrO_x oxide should be restricted to a certain value (ca. 4–8 wt.%) to elevate the catalytic stability of catalyst and meanwhile avoid a substantial decrease of the CO_2 conversion.

Thanks again to the reviewer for the valuable comments. Considering the reviewer's concern,

these relevant descriptions and discussion have been supplemented and above points were strengthened in the revised manuscript (Pages 26–28).

Q10: If the authors address satisfactorily the previous points, publication would be recommended.

Response: Thanks again to the reviewer for the very positive comments as well as the informative and instructive revision advices. With the help of these advices, the revised manuscript was improved greatly. In addition, as the reviewer can see, all the raised issues have been carefully and pertinently addressed. In particular, more characterizations about the used InZrO_x oxide and H-Beta zeolite have been provided. The dispersion of silica species on the surface of InZrO_x oxide was further evaluated by Aberration-corrected HAADF-STEM images, HR-TEM images and EDX-elemental mapping. The influence of co-feeding water on the catalytic performance and the surface silica protection mechanism were systemically investigated. Moreover, the interaction between the surface silica species and InZrO_x oxide and its relation to the catalytic stability of InZrO_x-Beta in the CO₂ hydrogenation have been well clarified.

We hope that our revision work is satisfactory to the reviewer and the acceptance of this manuscript for publication can be approved.

Thanks to the reviewers for the informative and constructive advices and kind efforts in handling this manuscript. As the reviewers can see, with the help of these advices, the revised manuscript was improved greatly; we would be very grateful if the acceptance of the revised manuscript can be approved.

REVIEWERS' COMMENTS

Reviewer #1 (Remarks to the Author):

The authors have satisfactorily addressed the reviewers' comments and improved the quality of this paper. I would recommend accepting it.

**Response to the reviewers' comments for the manuscript
NCOMMS-22-46197A**

To Reviewer #1:

Q0: The authors have satisfactorily addressed the reviewers' comments and improved the quality of this paper. I would recommend accepting it.

Response: Thanks a lot to the reviewer for the positive comments as well as the affirmation of our revision work.

Thanks again to the reviewers for the informative and constructive advices and kind efforts in handling this manuscript. These advices do help a lot for us to improve current manuscript.